# Provably Mitigating Overoptimization in RLHF: Your SFT Loss is Implicitly an Adversarial Regularizer

**Zhihan Liu**[1][*]   **Miao Lu**[2][*]   **Shenao Zhang**[1]   **Boyi Liu**[3]
**Hongyi Guo**[1]   **Yingxiang Yang**[3]   **Jose Blanchet**[2]   **Zhaoran Wang**[1]
[1]Northwestern University   [2]Stanford University   [3]ByteDance Inc.
{zhihanliu2027,shenaozhang2028,hongyiguo2025}@u.northwestern.edu
{miaolu, jose.blanchet}@stanford.edu, zhaoranwang@gmail.com
{boyi.liu01, yingxiang.yang}@bytedance.com

## Abstract

Aligning generative models with human preference via RLHF typically suffers from overoptimization, where an imperfectly learned reward model can misguide the generative model to output undesired responses. We investigate this problem in a principled manner by identifying the source of the misalignment as a form of distributional shift and uncertainty in learning human preferences. To mitigate overoptimization, we first propose a theoretical algorithm that chooses the best policy for an adversarially chosen reward model; one that simultaneously mini-mizes the maximum likelihood estimation of the loss and a reward penalty term. The penalty term is introduced to prevent the policy from choosing actions with spurious high proxy rewards, resulting in provable sample efficiency of the algo-rithm under a *partial coverage* style condition. Moving from theory to practice, the proposed algorithm further enjoys an equivalent but surprisingly easy-to-implement reformulation. Using the equivalence between reward models and the correspond-ing optimal policy, the algorithm features a simple objective that combines: (i) a preference optimization loss that directly aligns the policy with human preference, and (ii) a supervised learning loss that explicitly imitates the policy with a (suitable) baseline distribution. In the context of aligning large language models (LLM), this objective fuses the direct preference optimization (DPO) loss with the supervised fine-tuning (SFT) loss to help mitigate the overoptimization towards undesired responses, for which we name the algorithm Regularized Preference Optimization (RPO). Experiments of aligning LLMs demonstrate the improved performance of RPO compared with DPO baselines. Our work sheds light on the interplay between preference optimization and SFT in tuning LLMs with both theoretical guarantees and empirical evidence.

## 1 Introduction

A key step in building state-of-the-art LLMs is Reinforcement Learning from Human Feedback (RLHF) [12, 87], which aligns pretrained LLMs with human preferences using human assessment data, making the model more helpful, truthful, and harmless [42, 10]. Typically, RLHF first learns a reward model from data (pair-wise comparisons of responses) to quantify the human preferences of LLM outputs. Then it fine-tunes the LLM to maximize the learned reward using RL techniques.

In this pipeline, a crucial challenge is *reward overoptimization* or *reward hacking* [40, 62, 25]. Since the reward model is learned from finite data, it might not be perfectly aligned with the underlying human preference. Optimizing the LLM towards such an imperfectly learned and potentially overfitted

---

[*]Equal contribution.

38th Conference on Neural Information Processing Systems (NeurIPS 2024).

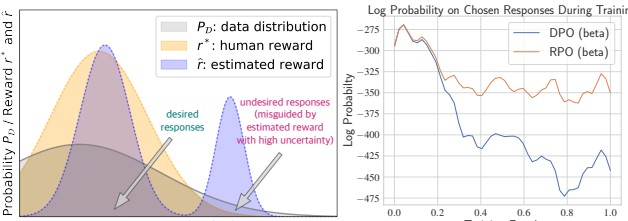

Figure 1: *Left*: Reward overoptimization due to the distributional shift and uncertainty in reward. *Right:* Overoptimization causes the probability of outputting preferred responses in the preference data to decrease substantially using original DPO proposed by [46]. Our algorithm (RPO) significantly alleviates this decrease. See more discussions in Section 6.

reward model leads to performance degeneration and a substantial decrease in the probability of choosing the preferred responses in the data [26, 45]. Given the importance of RLHF and the outlined challenge, a crucial research question is: *How to mitigate reward overoptimization in RLHF in a principled and efficient manner for better alignment?*

To answer the question, we model RLHF as an offline contextual bandit [42] and ascribe overoptimization to distributional shifts and reward uncertainty. Intuitively, when fine-tuning an LLM, the response (action) distribution of the tuned LLM could deviate from that of the training data. For the out-of-distribution responses, which are dissimilar with (or not well covered by) the responses in the data, the high inherent uncertainty of underlying human preferences could make the learned reward model misleading for out-of-distribution responses. In this situation, reward overoptimization can occur because the LLM is fine-tuned towards maximizing a reward model with defective out-of-distribution prediction, giving a potential consequence that the LLM responses are favored by the learned reward but less preferred by a human [86]. We illustrate these types of distributional shift and reward uncertainty issues inherent to overoptimization in Figure 1.

In this paper, we propose a new RLHF algorithm to mitigate reward overoptimization. From a high level, our theoretical algorithm seeks the best LLM for an *adversarially* chosen reward model that minimizes the sum of its maximum likelihood estimation loss and its own expected reward value. Intuitively, since the reward value is also minimized when minimizing the sum, it can automatically prevent the misleadingly high reward caused by the uncertainty inherent in having access to finite data. Furthermore, we show that the theoretical algorithm enjoys an easy implementation: it simply adopts a supervised fine-tuning (SFT) loss as a regularizer during training. By explicitly regularizing the LLM to imitate high-quality responses (e.g., preferred responses in dataset), the algorithm can effectively mitigate the issue of overoptimization. We establish theoretical guarantees and conduct experiments to demonstrate our findings, which we summarize next.

## 1.1 Our Contributions and Related Works

We summarize our contributions in three areas as follows.

**A theoretical algorithm under general function approximation.** Our first contribution is a new theoretical algorithm (Algorithm 1). It features an *unconstrained maximin* problem, outputting the optimal policy (LLM) against an adversarially chosen reward model that minimizes the summation of: (a) the MLE loss for estimating the underlying reward; and (b) a reward expected value term as a penalty that aims to prevent spuriously high reward estimation caused by data uncertainty and insufficient coverage. Algorithm 1 is compatible with general function approximations of the reward model, meaning that we do not impose any specific structural form to the hypothesis class of reward, demonstrating its generality, especially in language modeling.

In this regime of reward class, we establish the finite-sample suboptimality gap of Algorithm 1 as $\widetilde{\mathcal{O}}(C_{\text{coverage}}^2 \sqrt{\mathcal{N_R}/N})$ when competing with any LLM in terms of the underlying true human reward (Theorem 5.3). Here $N$ is the number of human comparison data, $\mathcal{N_R}$ is the complexity of the reward model class $\mathcal{R}$, and $C_{\text{coverage}}$ characterizes the coverage of the preference dataset with respect to the response distribution of the LLM to compete (please see Assumption 5.2 for details). This indicates that, as long as the training data well cover the LLM $\pi$ to compete, the algorithm is guaranteed to align an LLM to output responses as good as $\pi$ in terms of human reward, without suffering from overoptimization caused by distributional shifts and inherent uncertainty in human preference.

**An easy-to-implement practical objective.** Moving towards practice, we show that the objective of Algorithm 1 adopts a surprisingly simple and equivalent form for its use in practice. Specifically, with mild regularity conditions, we prove that the *maximin* objective (Algorithm1) is equivalent to the

corresponding *minimax* objective, which is further reduced to a single minimization problem for the reward model since its inner problem adopts a closed form solution. Inspired from recent progress in RLHF that explores reward-model-free methods to align LLMs [46], we further re-parameterize the reward model via its corresponding KL-regularized optimal policy. Then the minimization objective of the reward modeling naturally translates to a target for directly aligning the LLM, which we call Regularized Preference Optimization (RPO; Algorithm 2). The objective of RPO features a simple weighted combination of two losses:

$$\text{RPO objective } = \text{ Preference optimization loss } + \text{ Imitation (SFT) loss}.$$

Here the Preference optimization loss coincides with the DPO [46] objective, tending to optimize the LLM towards maximizing the underlying true reward. The Imitation (SFT) loss explicitly supervises the LLM to mimic the responses from a proper distribution well covered by the dataset. The choice of the distribution is guided and justified by our theory of Algorithm 1, but can also be flexibly adapted in practice, e.g., the preferred response in the dataset, or the responses of the initial model.

We highlight that the Imitation (SFT) loss serves as an important term to mitigate overoptimization. Even though the original DPO objective has already involved a KL regularization between the tuned LLM and the initial LLM, is not enough to prevent overoptimization. As we elaborate in Section 4, the KL-regularization weight of the DPO objective could only control the scale of the gradient per training example, while the RPO objective can further modify the gradient direction. Calling back to the theoretical Algorithm 1, such a modification of gradient direction originates from the reward penalty in the adversarial objective for the reward model. This modification, as we expose in our theoretical analysis, helps to mitigate overoptimization. *Thus, incoporating SFT loss in RLHF gives you a regularizer that provably mitigates overoptimization.*

**Empirical evaluations.** Following the training setup of two series of released chat models `Zephyr-7b-beta` (trained on the Ultrafeedback dataset [16] by DPO) and `Zephyr-7b-gemma` (trained on the Argilla-DPO-Mix-7K dataset [3] by DPO) [64], we implement RPO for the beta series and gemma series respectively to show that: (i) RPO is a flexible plug-in module and can be applied to different reference models. (ii) RPO can alleviate the overoptimization issue. (iii) RPO consistently achieves better alignment performance than DPO in in-data distribution. (iv) RPO can also achieve consistently better performance in standard LLM benchmarks like MT-bench and AlpacaEval 2.0, which shows its potential of mitigating overoptimization for better alignment performance, justifying our theory.

**Related works.** Due to space limitation, we refer the readers to Appendix A for a detailed discussion.

## 2 Preliminaries of RLHF

In this section, we introduce the mathematical framework of studying RLHF for aligning LLMs. We adopt the framework of offline contextual bandits [42], where we identify the context space $\mathcal{X}$ as the space of prompts and the action space $\mathcal{A}$ as the space of responses. An LLM, defined as a policy $\pi(\cdot|\cdot) : \mathcal{X} \mapsto \Delta(\mathcal{A})$, takes a prompt $x \in \mathcal{X}$ as input and output a response $a \in \mathcal{A}$ from $a \sim \pi(\cdot|x)$.

**Preference model.** Given any reward function $r : \mathcal{X} \times \mathcal{A} \mapsto \mathbb{R}$ belonging to certain reward class $\mathcal{R}$ that represents the "human's ratin" of LLM responses given prompts, we consider the Bradley-Terry model [9] of human preference. That is, given a prompt $x \in \mathcal{X}$ and two responses $a^1, a^0 \in \mathcal{A}$, the probability of $a^1$ being preferred to $a^0$ (denoted by $y = 1$, otherwise $y = 0$) is given by

$$\mathbb{P}_r(y = 1|x, a^1, a^0) = \frac{\exp(r(x, a^1))}{\exp(r(x, a^1)) + \exp(r(x, a^0))} = \sigma\big(r(x, a^1) - r(x, a^0)\big), \quad (2.1)$$

where $\sigma(z) = 1/(1 + \exp(-z))$ is the sigmoid function. For simplicity of future discussion, we explicitly write out the dependence of the preference probability $\mathbb{P}_r(\cdot)$ on the reward model $r \in \mathcal{R}$. In the section of theory, i.e., Section 5, we specify the assumptions on the reward model class $\mathcal{R}$.

**Learning protocol.** Typically, the RLHF pipeline starts from certain reference policy $\pi^{\text{ref}}$ obtained from pretraining. Then RLHF aligns the LLM based on certain human preference data. In this work, we consider offline RLHF setup, where the LLM is aligned using a fixed offline preference dataset $\mathcal{D}$. It consists of $N$ i.i.d. tuples in the form of $\mathcal{D} = \{(x_i, a_i^1, a_i^0, y_i)\}_{i=1}^N$. Here the prompt $x_i$ and the responses $a_i^1, a_i^0$ are distributed according to: $(x, a^1, a^0) \sim \mu_{\mathcal{D}}(\cdot)$, and conditioning on $(x_i, a_i^1, a_i^0)$, $y_i$ is distributed according to (2.1) for an underlying true (but unknown) reward model $r^\star \in \mathcal{R}$.

---
**Algorithm 1** Theoretical Algorithm: Maximin Objective
---
1: **Input**: Preference dataset $\mathcal{D}$, parameters $\beta, \eta > 0$, reference policy $\pi^{\text{ref}}$, baseline policy $\pi^{\text{base}}$.
2: **Output**: Policy $\widehat{\pi}$ given by (3.2) with the cross-entropy loss function $\mathcal{L}_\mathcal{D}$ defined in (3.1)..
---

**Performance metric.** The target of RLHF is to align an LLM, or equivalently, to learn a policy $\pi$, so as to maximize the expected true reward $r^\star$. Thus, we define the value function of any policy $\pi$ as

$$J(\pi) = \mathbb{E}_{x \sim d_0, a \sim \pi(\cdot|x)}\big[r^\star(x, a)\big]. \tag{2.2}$$

Here we allow the prompt distribution $d_0(\cdot)$ to be different from that of the offline dataset distribution $\mu_\mathcal{D}(\cdot)$, but is assumed to be known. In the meanwhile, we consider the policies that share the same support as the reference policy $\pi^{\text{ref}}$ [71], that is, we take a policy class $\Pi$ as

$$\Pi = \Big\{\pi : \mathcal{X} \mapsto \Delta(\mathcal{A}) \,\Big|\, \text{Supp}(\pi(\cdot|x)) \subseteq \text{Supp}(\pi^{\text{ref}}(\cdot|x)), \ \forall x \in \mathcal{X}\Big\}. \tag{2.3}$$

The performance gap of a learned policy $\widehat{\pi} \in \Pi$ w.r.t. any other policy $\pi \in \Pi$ is measured as

$$\text{Gap}^\pi(\widehat{\pi}) = J(\pi) - J(\widehat{\pi}), \ \textit{given policy } \pi. \tag{2.4}$$

The goal is to propose a sample-efficient and also implementation-friendly algorithm to learn a policy $\widehat{\pi} \in \Pi$ able to compete with any given policy $\pi \in \Pi$ in terms of $\text{Gap}^\pi(\widehat{\pi}) \leq \varepsilon$, with sample complexity polynomial in $1/\varepsilon$ and logarithmic in the complexity of $\mathcal{R}$.

## 3 A Theory-motivated Objective

Our method seeks to find the best policy $\widehat{\pi}$ against an *adversarially* chosen reward model $\widehat{r}_{\text{adv}}$ that minimizes a weighted sum of its expected value and the maximum likelihood estimation (MLE) loss. Intuitively, such a reward model can prevent the overoptimization issue by taking its own value into account when minimizing the MLE loss. Since the reward value is also minimized when minimizing the sum, this method prevents the misleadingly high reward caused by the uncertainty due to finite data. Formally, given two hyperparameters $\beta, \eta > 0$ and a "baseline policy" $\pi^{\text{base}}$, we define

$$T_{\beta,\eta}^{\text{adv}}(\pi) = \min_{r \in \mathcal{R}} \left\{ \eta \mathbb{E}_{\substack{x \sim d_0, a^1 \sim \pi(\cdot|x), \\ a^0 \sim \pi^{\text{base}}(\cdot|x)}} \Big[r(x, a^1) - r(x, a^0) - \beta \cdot \text{KL}\big(\pi(\cdot|x)\|\pi^{\text{ref}}(\cdot|x)\big)\Big] + \mathcal{L}_\mathcal{D}(r) \right\},$$

where the loss function $\mathcal{L}_\mathcal{D}(\cdot)$ is the average negative log-likelihood function of the BT model (2.1) (and here it becomes the cross-entropy loss) over the preference dataset $\mathcal{D}$, defined as

$$\mathcal{L}_\mathcal{D}(r) = -\widehat{\mathbb{E}}_\mathcal{D}\left[y_i \log\big(\sigma\big(r(x_i, a_i^1) - r(x_i, a_i^0)\big)\big) + (1 - y_i) \log\big(\sigma\big(r(x_i, a_i^0) - r(x_i, a_i^1)\big)\big)\right]. \tag{3.1}$$

As we can see, $T_{\beta,\eta}^{\text{adv}}(\pi)$ is the minimum value of a weighted sum of the MLE loss and the expected reward value of $\pi$, but with two important modifications that we explain in the following.

*Firstly*, we subtract another expected reward of certain policy $\pi^{\text{base}}$. This is because the BT model (2.1) essentially only uses the reward differences to define the preference probabilities. As a result, the data can only reveal information of the differences between the true reward $r^\star$ of different responses [78]. Accordingly, we subtract such a baseline expected reward value to match this observation. The choice of the baseline policy is discussed in the theory part (Section 5) and experiments (Section 6).

*Secondly*, we subtract a KL divergence between $\pi$ and $\pi^{\text{ref}}$ from the expected reward, weighted by the coefficient $\beta > 0$. Such a term is for practical considerations that would be explained in Sections 4 and 5.2. We note that the KL regularized reward is commonly adopted in RLHF practice to ensure the learned policy is not far away from the reference policy [42, 71].

Finally, the overall algorithm design (Algorithm 1) is to output the policy that maximizes $T_{\beta,\eta}^{\text{adv}}(\pi)$, i.e., $\widehat{\pi} \in \arg\max_{\pi \in \Pi} T_{\beta,\eta}^{\text{adv}}(\pi)$, which gives the following theoretical target:

$$\widehat{\pi} \in \underset{\pi \in \Pi}{\operatorname{argmax}} \min_{r \in \mathcal{R}} \left\{ \eta \mathbb{E}_{\substack{x \sim d_0, a^1 \sim \pi(\cdot|x), \\ a^0 \sim \pi^{\text{base}}(\cdot|x)}} \left[ r(x, a^1) - r(x, a^0) - \beta \cdot \text{KL}\big(\pi(\cdot|x) \| \pi^{\text{ref}}(\cdot|x)\big) \right] + \mathcal{L}_{\mathcal{D}}(r) \right\}.$$
(3.2)

Given the form of (3.2), we name it the *maximin* objective in the sequel. Upon seeing (3.2), one might be arguing that such a theory-motivated objective seems hard to implement in practice. Nevertheless, in the coming Section 4, we demonstrate that the maximin objective (3.2) adopts an easy-to-implement equivalent form, allowing us to design a practical algorithm for aligning LLMs.

# 4 An Equivalent and Implementation-friendly Objective

In this section, we propose another *minimax*-style objective that is equivalent to the maximin objective (3.2). Based on the minimax objective, we propose a new LLM aligning algorithm called Regularized Preference Optimization (RPO). It draws inspirations from the reparametrization technique originated in Direct Preference Optimization (DPO) [46] and goes beyond to further address the overoptimization issue in offline RLHF by incorprating an SFT loss as an explicit adversarial regularizer.

**An equivalent minimax objective.** If the reward model class $\mathcal{R}$ satisfies certain regularity conditions, which we discuss in detail in Section 5.2, the minimax theorem holds: solving the *maximin* objective (3.2) is *equivalent* to solving a *minimax* target, given by

$$\min_{r \in \mathcal{R}} \max_{\pi \in \Pi} \left\{ \eta \mathbb{E}_{\substack{x \sim d_0, a^1 \sim \pi(\cdot|x), \\ a^0 \sim \pi^{\text{base}}(\cdot|x)}} \left[ r(x, a^1) - r(x, a^0) - \beta \cdot \text{KL}\big(\pi(\cdot|x) \| \pi^{\text{ref}}(\cdot|x)\big) \right] + \mathcal{L}_{\mathcal{D}}(r) \right\}. (4.1)$$

Such a minimax formulation (4.1) is the starting point of our practical algorithm. The magic of (4.1) is that the inner maximization problem adopts a closed form solution, which further simplifies such an objective. To see this, note that given any reward model $r \in \mathcal{R}$, the inner problem is equivalent to

$$\max_{\pi \in \Pi} \left\{ \mathbb{E}_{x \sim d_0, a \sim \pi(\cdot|x)} \left[ r(x, a) - \beta \cdot \text{KL}\big(\pi(\cdot|x) \| \pi^{\text{ref}}(\cdot|x)\big) \right] \right\}. \tag{4.2}$$

It has been well established that the policy that maximizes the KL-regularized expected reward (4.2) has a closed form solution. Due to its importance, we present it as the following lemma.

**Lemma 4.1** (Oracle optimal KL-regularized policy). *Given any reward model $r \in \mathcal{R}$, the optimal policy $\pi_r$ to the maximization problem* (4.2) *is given by*

$$\pi_r(\cdot|x) = \frac{1}{Z_r(x)} \cdot \pi^{\text{ref}}(\cdot|x) \cdot \exp\big(\beta^{-1} r(x, \cdot)\big), \ Z_r(x) = \int_{a \in \mathcal{A}} \exp\big(\beta^{-1} r(x, a)\big) \, \mathrm{d}\pi^{\text{ref}}(a|x),$$

*and correspondingly the optimal value of* (4.2) *is given by* (4.2) $= \mathbb{E}_{x \sim d_0}[\beta \cdot \log(Z_r(x))]$.

Specifically, by Lemma 4.1, we can solve the inner maximization problem in (4.1) and obtain that

$$(4.1) = \min_{r \in \mathcal{R}} \left\{ \eta \mathbb{E}_{x \sim d_0, a^0 \sim \pi^{\text{base}}(\cdot|x)} \left[ -r(x, a^0) + \beta \cdot \log\big(Z_r(x)\big) \right] + \mathcal{L}_{\mathcal{D}}(r) \right\}.$$

Furthermore, from Lemma 4.1, one immediately see that given any reward model $r \in \mathcal{R}$, we can reparameterize it via its corresponding optimal KL-regularized policy $\pi_r$ [46], that is,

$$r(x, \cdot) = \beta \cdot \log\left(\frac{\pi_r(\cdot|x)}{\pi^{\text{ref}}(\cdot|x)}\right) + \beta \cdot \log(Z_r(x)). \tag{4.3}$$

Taking (4.3) back into (4.1), we are able to further simplify it as

$$(4.1) = \min_{r \in \mathcal{R}} \left\{ \eta \mathbb{E}_{x \sim d_0, a^0 \sim \pi^{\text{base}}(\cdot|x)} \left[ -\beta \cdot \log(\pi_r(a^0|x)) \right] + \mathcal{L}_{\mathcal{D}}\left(\beta \cdot \log\left(\frac{\pi_r(\cdot|\cdot)}{\pi^{\text{ref}}(\cdot|\cdot)}\right)\right) \right\}. \tag{4.4}$$

Thanks to the KL-regularization term in the original minimax objective (4.1) (or equivalently, the maximin objective (3.2)), we have the following theorem. It theoretically shows that the policy $\pi_{\widehat{r}}$ associated with the reward model $\widehat{r}$ solving (4.4) also solves the maximin target (3.2) of the theoretical algorithm (Algorithm 1) that enjoys finite-sample convergence guarantees. (Please see Section 5.2 for a formal statement and proof of Theorem 4.2).

---

**Algorithm 2** Practical Algorithm: Regularized Preference Optimization (RPO)

---
1: **Input**: Preference dataset $\mathcal{D}$, parameters $\beta, \eta > 0$, reference policy $\pi^{\text{ref}}$, baseline policy $\pi^{\text{base}}$.
2: **Output**: Policy $\pi_{\widehat{\theta}}$ obtained by optimizing objective (4.5).

---

**Theorem 4.2** (Equivalence between *maximin* and *minimax* algorithm (informal)). *Under certain regularity assumptions on $\mathcal{R}$ and given $\eta, \beta > 0$, solving the minimax objective* (4.1) *via* (4.4), *i.e.,*

$$\widehat{r} = \underset{r \in \mathcal{R}}{\operatorname{argmin}} \left\{ \eta \mathbb{E}_{x \sim d_0, a^0 \sim \pi^{\text{base}}(\cdot|x)} \Big[ -\beta \cdot \log(\pi_r(a^0|x)) \Big] + \mathcal{L}_{\mathcal{D}} \left( \beta \cdot \log \left( \frac{\pi_r(\cdot|\cdot)}{\pi^{\text{ref}}(\cdot|\cdot)} \right) \right) \right\},$$

*then the corresponding optimal KL-regularized policy $\pi_{\widehat{r}}$ also solves the maximin objective* (3.2).

**Regularized Preference Optimization.** Target (4.4) gives a quite simple objective to use in practice! Since (4.4) depends on $r \in \mathcal{R}$ only through its corresponding optimal policy $\pi_r$, one can formulate a minimization objective over a parameterized policy $\pi_\theta$, i.e., the LLM to be aligned, and directly optimize the parameters $\theta \in \Theta$. More formally, the new RLHF objective becomes

$$\min_{\theta \in \Theta} \left\{ \mathcal{L}_{\text{RPO}}(\theta) := \eta\beta \cdot \underbrace{\mathbb{E}_{x \sim d_0, a^0 \sim \pi^{\text{base}}(\cdot|x)} \Big[ -\log(\pi_\theta(a^0|x)) \Big]}_{\text{Imitation (SFT) loss}} + \underbrace{\mathcal{L}_{\mathcal{D}} \left( \beta \cdot \log \left( \frac{\pi_\theta(\cdot|\cdot)}{\pi^{\text{ref}}(\cdot|\cdot)} \right) \right)}_{\text{Preference opt. loss}} \right\}. \tag{4.5}$$

In (4.5), the second term coincides with the objective of DPO algorithm [46] which optimizes the policy towards maximizing the underlying true reward, and the first term stands for a regularization term weighted by $\eta \cdot \beta$ which *explicitly* regularizes the policy to imitate the baseline policy. Therefore, we name the resulting algorithm as Regularized Preference Optimization (RPO). We summarize it abstractly in Algorithm 2. As for DPO, implementing RPO does not require to maintain a reward model $r$. Thus it is computationally more friendly compared to reward-based algorithms.

**How does RPO improve DPO?** We illustrate the effect of the imitation loss by analyzing the gradient of the RPO target $\mathcal{L}_{\text{RPO}}(\theta)$ in (4.5). Notice that by (4.5) we have

$$\nabla_\theta \mathcal{L}_{\text{RPO}}(\theta) = \eta\beta \cdot \underbrace{\mathbb{E}_{x \sim d_0, a^0 \sim \pi^{\text{base}}(\cdot|x)} \Big[ -\nabla_\theta \log(\pi_\theta(a^0|x)) \Big]}_{\text{increase the alignment with the baseline policy}} + \underbrace{\nabla_\theta \mathcal{L}_{\text{DPO}}(\theta)}_{\text{decrease the DPO Loss}},$$

where the derivative of the DPO loss $\nabla_\theta \mathcal{L}_{\text{DPO}}(\theta)$ is given by the following,

$$\nabla_\theta \mathcal{L}_{\text{DPO}}(\theta) = -\widehat{\mathbb{E}}_{\mathcal{D}} \bigg[ \underbrace{\beta \cdot \sigma\big(\widehat{r}_\theta(x, a_{\text{rej}}) - \widehat{r}_\theta(x, a_{\text{cho}})\big)}_{\text{gradient weight}} \cdot \Big( \nabla_\theta \log \pi_\theta(a_{\text{cho}}|x) - \nabla_\theta \log \pi_\theta(a_{\text{rej}}|x) \Big) \bigg].$$

For simplicity we denote $\widehat{r}_\theta(x, a) = \beta \cdot \log(\pi_\theta(x, a)) / \log(\pi^{\text{ref}}(x, a))$, $a_{\text{cho}}$ for the chosen response and $a_{\text{rej}}$ for the rejected response. Intuitively, RPO (4.5) modifies the **gradient direction** of DPO to ensure the alignment with the baseline policy $\pi^{\text{base}}$, and the hyper-parameter $\eta$ controls the power of alignment. In comparison, the hyper-parameter $\beta$ in DPO only controls the **gradient weight** when increasing the likelihood of $a_{\text{cho}}$ and decreasing the likelihood $a_{\text{rej}}$. In this perspective, the hyper-parameter $\beta$ only changes the scale of the gradient instead of the direction. By introducing $\eta$, we stabilize the training and reduce the side-effect of uncertain labels in data to prevent overoptimization.

## 5   Theoretical Analysis

In this section, we establish theoretical analysis for Algorithms 1 and 2. We take the space of prompts and responses as compact subsets $\mathcal{X} \subseteq \mathbb{R}^{d_{\mathcal{X}}}$ and $\mathcal{A} \subseteq \mathbb{R}^{d_{\mathcal{A}}}$. We take the policy class $\Pi$ as (2.3).

### 5.1   Establishing the Sample Complexity of Maximin Objective (Algorithm 1)

**Assumption 5.1** (True reward model). *We assume that the true reward model $r^\star \in \mathcal{R}$, and for any $r \in \mathcal{R}$ and $(x, a) \in \mathcal{X} \times \mathcal{A}$, it holds that $r(x, a) \in [0, R]$.*

**Assumption 5.2** (Partial coverage coefficient [78]). *Given a policy $\pi \in \Pi$, the coverage coefficient of the offline dataset distribution $\mu_{\mathcal{D}}$ w.r.t. reward model class $\mathcal{R}$, policy $\pi$, and the baseline policy $\pi^{\mathrm{base}}$, denoted by $C_{\mu_{\mathcal{D}}}(\mathcal{R}; \pi, \pi^{\mathrm{base}})$, is defined as*

$$
\max\left\{0, \sup_{r \in \mathcal{R}} \frac{\mathbb{E}_{x \sim d_0, a^1 \sim \pi(\cdot|x), a^0 \sim \pi^{\mathrm{base}}(\cdot|x)}\left[(r^\star(x, a^1) - r^\star(x, a^0)) - (r(x, a^1) - r(x, a^0))\right]}{\sqrt{\mathbb{E}_{(x, a^1, a^0) \sim \mu_{\mathcal{D}}}\left[\left|(r^\star(x, a^1) - r^\star(x, a^0)) - (r(x, a^1) - r(x, a^0))\right|^2\right]}}\right\}.
$$

*We assume that $C_{\mu_{\mathcal{D}}}(\mathcal{R}; \pi, \pi^{\mathrm{base}}) < +\infty$ for the policy $\pi$ to compete. We remark that the quantity $C_{\mu_{\mathcal{D}}}(\mathcal{R}; \pi, \pi^{\mathrm{base}})$ is upper bounded by the density ratio $\|d_0 \otimes \pi \otimes \pi^{\mathrm{base}}/\mu_{\mathcal{D}}\|_\infty$.*

Assumption 5.1 is standard in sample complexity analysis [85, 78, 74]. Assumption 5.2 characterizes how well the dataset $\mathcal{D}$ covers the policy $\pi$ to compete. To achieve provable sample efficiency, we only require that $\mathcal{D}$ covers the target policy $\pi$, a weak partial coverage style assumption for theoretical analysis. To illustrate it, when calling back to Figure 1, the data distribution therein well covers those nearly optimal responses under $r^\star$, but does not sufficiently cover the responses with low $r^\star$.

Under such a partial coverage data condition, however, human preference of responses $a \in \mathcal{A}$ that are not well covered by the dataset $\mathcal{D}$ can be poorly estimated, misguiding the policy $\widehat{\pi}$ to behave suboptimally if it is overoptimized (recall Figure 1). Fortunately, the following theorem shows that Algorithm 1 provably mitigates the overoptimization issue and achieves a finite-sample convergence of the suboptimality gap (2.4) competing with $\pi$. Proof is in Appendix D.

**Theorem 5.3** (Suboptimality of Algorithm 1). *Taking the policy class $\Pi$ as (2.3), supposing that Assumptions 5.1 and 5.2 hold, and assuming that the reward model class $\mathcal{R}$ has a finite $\varepsilon$-epsilon covering number under $\|\cdot\|_\infty$-norm $\mathcal{N}_\varepsilon(\mathcal{R}, \|\cdot\|_\infty) < +\infty$ with $\varepsilon = (6 \cdot (1 + e^R) \cdot N)^{-1}$. Setting*

$$
\eta = (1 + \exp(R))^{-2} \cdot \sqrt{24 \log\left(\mathcal{N}_\varepsilon(\mathcal{R}, \|\cdot\|_\infty)/\delta\right)/N}, \quad \beta = 1/\sqrt{N}
$$

*in Algorithm 1. Then the output policy $\widehat{\pi}$ of Algorithm 1 satisfies that with probability at least $1 - \delta$,*

$$
\mathrm{Gap}^\pi(\widehat{\pi}) \leq \frac{\sqrt{6}\left(1 + \exp(R)\right)^2\left(\left(C_{\mu_{\mathcal{D}}}(\mathcal{R}; \pi, \pi^{\mathrm{base}})\right)^2 + 1\right)\iota + 4\mathbb{E}_{x \sim d_0}\left[\mathrm{KL}\left(\pi(\cdot|x)\|\pi^{\mathrm{ref}}(\cdot|x)\right)\right]}{4\sqrt{N}},
$$

*where $\iota = \sqrt{\log\left(\mathcal{N}_\varepsilon(\mathcal{R}, \|\cdot\|_\infty)/\delta\right)}$ with $\varepsilon = (6 \cdot (1 + e^R) \cdot N)^{-1}$. Here, $N$ denotes the number of preference pairs in $\mathcal{D}$, $R$ denotes the upper bound of the reward models, and the partial coverage coefficient $C_{\mu_{\mathcal{D}}}(\mathcal{R}; \pi, \pi^{\mathrm{base}})$ is defined in Assumption 5.2.*

**Remark 5.4** (Choice of the baseline policy). *As is indicated by Assumption 5.2, the least requirement is that $\pi^{\mathrm{base}}$ can be covered by the offline data distribution. E.g., we can take $\pi^{\mathrm{base}}$ as the distribution of the preferred responses in the data. In this case, the SFT loss in RPO explicitly regularizes the LLM to imitate the preferred responses. We choose this type of baseline policy in our experiments.*

## 5.2 Equivalence between Maximin and Minimax Objectives

Now we formally show that the theoretical target (maximin objective (3.2)) and the target for practical algorithm design (minimax objective (4.1)) are equivalent under certain regularity conditions. This can naturally extend the sample complexity of Algorithm 1 (Section 5.1) to that of minimax-based algorithms in Section 4, providing the theoretical guarantee for our practical algorithm design (RPO).

First, for notational simplicity, we denote the optimization target we investigate in Sections 3 and 4 as

$$
\phi(\pi, r) := \eta \cdot \mathbb{E}_{\substack{x \sim d_0, a^1 \sim \pi(\cdot|x) \\ a^0 \sim \pi^{\mathrm{base}}(\cdot|x)}}\left[r(x, a^1) - r(x, a^0) - \beta \cdot D_{\mathrm{KL}}\left(\pi(\cdot|x)\|\pi^{\mathrm{ref}}(\cdot|x)\right)\right] + \mathcal{L}_{\mathcal{D}}(r), \quad (5.1)
$$

for any $(\pi, r) \in \Pi \times \mathcal{R}$. Our result relies on the following assumptions on the reward model class $\mathcal{R}$.

**Assumption 5.5** (Regularity of reward model class). *We assume the following things on the reward model class $\mathcal{R}$: (i) the space $\mathcal{R}$ is a compact topological space; (ii) the function $\phi$ in (5.1) is convex-like on $\mathcal{R}$, that is, for any $r_1, r_2 \in \mathcal{R}$ and $\alpha \in [0, 1]$, there exists $r_3 \in \mathcal{R}$ such that*

$$
\phi(\pi, r_3) \leq \alpha \cdot \phi(\pi, r_1) + (1 - \alpha) \cdot \phi(\pi, r_2), \quad \forall \pi \in \Pi, \quad (5.2)
$$

We note if $\mathcal{R}$ is convex, e.g., a linear model class [85, 71, 86] or more general the Lipschitz continuous model class $\mathcal{R}$, we can directly obtain that the function $\phi(\pi, \cdot)$ is *convex* over $\mathcal{R}$ (since the dependence on $r \in \mathcal{R}$ is linear terms plus a convex loss $\mathcal{L}_\mathcal{D}$ of $r \in \mathcal{R}$), which implies the convex-like property (5.2). Under Assumption 5.5, it holds that (Lemma E.1)

$$\max_{\pi \in \Pi} \min_{r \in \mathcal{R}} \phi(\pi, r) = \min_{r \in \mathcal{R}} \max_{\pi \in \Pi} \phi(\pi, r). \tag{5.3}$$

Furthermore, thanks to the KL-divergence regularization in $\phi$ which intuitively makes $\phi$ "strongly concave" over the policy $\pi$, (5.3) can gives us the following stronger result, proved in Appendix E.1.

**Theorem 5.6** (Formal statement of Theorem 4.2). *For the policy class $\Pi$ defined in* (2.3) *and the reward model class $\mathcal{R}$ satisfying Assumption 5.5, consider the following policy defined as*

$$\pi_{\widehat{r}} \in \operatorname*{argmax}_{\pi \in \Pi} \phi(\widehat{r}, \pi), \quad \text{where} \quad \widehat{r} \in \operatorname*{argmin}_{r \in \mathcal{R}} \max_{\pi \in \Pi} \phi(\pi, r). \tag{5.4}$$

*Then the policy $\pi_{\widehat{r}}$ also satisfies the maximin objective* (3.2) *of Algorithm 1, that is,*

$$\pi_{\widehat{r}} \in \operatorname*{argmax}_{\pi \in \Pi} \min_{r \in \mathcal{R}} \phi(\pi, r).$$

Theorem 5.6 shows that the optimal KL-regularized policy associated with the reward model solving the minimax objective (3.2) also solves the maximin objective (i.e., objective (4.1) of Algorithm 1). This further allows us to extend our theoretical guarantee of Algorithm 1 (Section 5.1) to that of minimax-based algorithms, justifying our practical algorithm design in Section 4.

**Corollary 5.7** (Suboptimality of minimax-based algorithm). *Take the policy class $\Pi$ in* (2.3) *and the reward model class satisfying Assumption 5.5. Given any given policy $\pi$ to compete, if Assumption 5.2 holds for $\pi$, then under the same choice of $\eta$ and $\beta$ as in Theorem 5.3, the policy $\pi_{\widehat{r}}$ defined in* (5.4) *satisfies that* $\operatorname{Gap}^\pi(\pi_{\widehat{r}}) \leq \widetilde{\mathcal{O}}(1/\sqrt{N})$ *with probability at least $1 - \delta$.*

# 6 Experiments

In this section, we provide a detailed empirical analysis of RPO to highlight the following four key points: (1) RPO is a flexible plug-in module and can be applied to different reference models. (2) RPO can alleviate the overoptimization issue in the training phase by giving more trust to the chosen responses in the preference dataset. (3) As a justification of our theoretical analysis, RPO achieves better alignment performance than DPO in in-data distribution. (4) RPO can also achieve consistently better performance in LLM benchmarks like MT-bench [83] and AlpacaEval 2.0 [20], which shows the potential of mitigating overoptimization for better generalization performance. The code for the experiments can be found in https://github.com/YSLIU627/Regularized-Preference-Optimization/tree/master.

**Experiment setup.** To show that RPO is a flexible plug-in module regardless of the reference model, we follow the training setup for two well-studied series of released chat models with around 7 billion parameters trained by DPO: `Zephyr-7b-beta` and `Zephyr-7b-gemma` [64] to implement RPO in beta and gemma series. Mirrored by their training configurations, we introduce how we select the reference model and the preference dataset for our training on these two series as follows. For the beta series, we use `mistral-7b-sft-beta` as the reference model $\pi^{\text{ref}}$. `mistral-7b-sft-beta` is a fine-tuned version of `Mistral-7b-v0.1` on the distilled version of the UltraChat dataset [17], which contains approximately 200k examples of multi-turn dialogues generated by GPT-3.5-TURBO. For the training preference dataset, we use Ultrafeedback Dataset [16], which consists of approximately 60k prompts. For the gemma series, we use `zephyr-7b-gemma-sft-v0.1` as our reference model $\pi^{\text{ref}}$. `zephyr-7b-gemma-sft-v0.1` is a fine-tuned version of `gemma-7b` on the Deita dataset [35], which involves around 10k distilled SFT data. For the training preference dataset, we use Argilla-DPO-Mix-7K Dataset [3], which is a mixture of multiple distilled public preference datasets. For simplicity, we denote Ref. (beta) as the reference model, DPO (beta) as the model trained by DPO, RPO (beta) as the model trained by RPO, all for the beta series. We use the same notations for the gemma series.

**Practical implementation.** According to Algorithm 2 and as discussed in Remark 5.4, we implement RPO by adding an SFT loss (log probability of chosen responses in the preference dataset) to the original DPO loss. By comparing the evaluation performance on the test split of the training

dataset, we select the hyperparameter $\eta$ as $0.005$ for both RPO (beta) and RPO (gemma). During the training of DPO and RPO, We keep the remaining hyperparameters including $\beta$, batch size, and learning rate to be the same for a fair comparison. Please see Appendix F.1 for a detailed training configuration.

**RPO alleviates overoptimization.** As mentioned in the introduction part, DPO is observed to have a significant and continuous decrease in log probability on chosen responses [26, 45] during training and we regard it as the consequence of overoptimization. Implied by our theory, overoptimization could arise when the model maximizes its own proxy reward formed on the responses less covered by the data. Due to the overoptimization, the model tends to disprefer the chosen responses as they are away from the maximizers of the proxy reward despite that some chosen responses are highly preferred by humans. Consistent with our theoretical conclusion, we empirically find that RPO can indeed alleviate overoptimization in DPO. During the training phase of both beta and gemma series, we observe that the log probability given by the RPO-trained model is notably higher than that given by the DPO-trained model for the chosen responses, which are shown in Fig. 1 and 2.

**RPO improves the alignment ability in in-data distribution.** For the in-data distribution evaluation, we select the 200 prompts (which are not used in the selection of $\eta$) in the test split of the training dataset to let the reference model, DPO, and RPO generate the response respectively. We choose GPT-4 to annotate the preference in the response pairs. Though we instruct GPT-4 to give an annotation among win, lose, and, tie (please see the full prompt in Appendix F.2), GPT-4 may still give undesired annotations. Hence, we filter all the undesired annotations and collect 150 examples for evaluation. We report the pairwise win rate among Ref., RPO, and DPO in Table 1 for both the beta and gemma series. To show a more illustrative comparison between DPO and RPO, we provide the barplot to report the number of pairwise examples annotated by GPT-4 in Fig. 3 and Fig. 4. We observe that for both beta and gemma series, RPO has a better performance than DPO in terms of both RPO/DPO-SFT and RPO-DPO win rates. The performance improvement matches our theoretical results in Corollary 5.7, which shows the credit of the alleviation of overoptimization.

Table 1: Pairwise win rate (left vs. right) among RPO-trained model, DPO-trained model, and the reference model. Annotated by GPT-4, evaluations of beta and gemma series are made on the 150 examples of the test split of the Ultrafeedback and the Argilla-DPO-Mix-7K dataset, respectively.

| Win rate (%) | RPO (beta) | Ref. (beta) | DPO (beta) | Win rate (%) | RPO (gemma) | Ref. (gemma) | DPO (gemma) |
|---|---|---|---|---|---|---|---|
| RPO (beta) | 50.0 | **79.0** | **56.0** | RPO (gemma) | 50.0 | **71.7** | **54.0** |
| Ref. (beta) | 21.0 | 50.0 | 22.7 | Ref. (gemma) | 28.3 | 50.0 | 32.7 |
| DPO (beta) | 44.0 | 77.3 | 50.0 | DPO (gemma) | 46.0 | 67.3 | 50.0 |

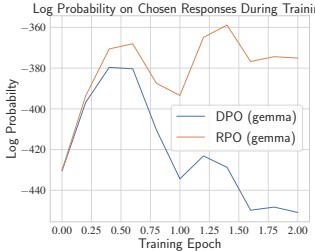

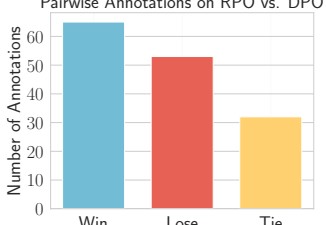

Figure 2: Log probability of the model for chosen responses during the training of RPO (gemma) and DPO (gemma).

Figure 3: Pairwise annotations (by GPT-4) on RPO (beta) vs. DPO (beta) on the test split of the Ultrafeedback dataset.

Figure 4: Pairwise annotations (by GPT-4) on RPO (gemma) vs. DPO (gemma) on the test split of the Argilla-DPO-Mix-7K dataset.

**RPO consistently improves the benchmark performance.** We further evaluate the reference model, RPO-trained model, DPO-trained model, and the officially released DPO-trained model for both beta and gemma series in two standard LLM chat benchmarks: MT-Bench and AlpacaEval 2.0. MT-Bench is a multi-turn benchmark that contains 160 questions across eight different domains of knowledge. The score for MT-Bench is evaluated by GPT-4 on a scale from 1 to 10. AlpacaEval 2.0 is a single-turn benchmark including 805 questions on different topics, mostly focused on helpfulness.

The metrics of AlpacaEval 2.0 are the win rate and Length-Control (LC) win rate compared with GPT-4 Preview (11/06), where the annotator is also GPT-4 Preview (11/06) and LC win rate is proposed to mitigate the length bias of GPT-4. The results are summarized in Table 2, which shows that RPO consistently exceeds the performance of all the competitors (DPO, Reference model, and the officially released model trained by DPO) on MT-Bench and AlpacaEval 2.0. We also provide additional results on the pairwise win rate for these two benchmarks in Appendix F.3 to illustrate the performance improvement. Finally, we remark that RPO is a flexible plug-in module and can steadily improve the benchmark performance without changing the original training configuration or accessing extra preference data. This also sheds light on the potential of mitigating overoptimization for better alignment and generalization performance.

Table 2: Results on MT-Bench scores and AlpacaEval 2.0. `zephyr-beta-7b` and `zephyr-gemma-7b` are the officially released models. win rates and Length-Control (LC) win rates in AlpacaEval 2.0 are evaluated by GPT-4 compared with GPT-4.

| Model Name | MT-Bench Score | AlpacaEval 2.0 | | Model Name | MT-Bench Score | AlpacaEval 2.0 | |
| --- | --- | --- | --- | --- | --- | --- | --- |
| | | LC win rate (%) | win rate (%) | | | LC win rate (%) | win rate (%) |
| RPO (beta) | **7.381** | **23.28** | **21.01** | RPO (gemma) | **7.916** | **15.51** | **13.85** |
| Ref. (beta) | 5.088 | 7.19 | 4.69 | Ref. (gemma) | 7.266 | 8.35 | 4.61 |
| DPO (beta) | 7.278 | 21.15 | 17.27 | DPO (gemma) | 7.688 | 15.36 | 13.69 |
| `zephyr-beta-7b` | 7.200 | 13.20 | 10.99 | `zephyr-gemma-7b` | 7.719 | 14.78 | 12.14 |

**RPO also improves the math, reasoning, and coding abilities.** In addition to the MT-Bench and AlpacaEval 2.0 benchmarks, we introduce more benchmarks on the math, reasoning, and coding tasks for evaluations of the RPO algorithm. Specifically, we choose the Grade School Math 8K (GSM8K) [14], AI2 Reasoning Challenge (ARC) [13], and Mostly Basic Python Programming (MBPP) [4] to measure math, reasoning, and coding abilities of the model trained by RPO, respectively. To due space limitation, we refer the readers to Appendix G for the setups and results of these experiments.

## 7 Conclusions

This paper proposes a new algorithm that provably mitigates reward overoptimization in RLHF. We establish its finite-sample convergence under a partial coverage style data condition, and provide an equivalent practical implementation, RPO. As a flexible plug-in module, RPO exhibits consistent improvement over the DPO baseline and effectively mitigates overoptimization. Future work includes extending our idea of algorithm design to online (iterative) RLHF where preference data are collected and updated iteratively during LLM fine-tuning. We give more detailed discussions in Appendix B.

## Acknowledgement

Zhaoran Wang acknowledges National Science Foundation (Awards 2048075, 2008827, 2015568, 1934931), Simons Institute (Theory of Reinforcement Learning), Amazon, J.P. Morgan, and Two Sigma for their supports. The authors thank Junyan Zhang on valuable discussions on the equivalence between the min-max and max-min optimization.

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

# A  Related Works

In the following, we relate our work to recent lines of RLHF research on both theory and practice sides. We also review related works on reward hacking and overoptimization in RLHF.

**RLHF: algorithm design.**  The technique of RLHF [12, 87, 42, 6, 18, 55] has recently demonstrated its great importance in building the state-of-the-art LLMs, including ChatGPT [1], Gemini [61], Claude [2]. In the RLHF pipeline, the LLM is fine-tuned towards maximizing a learned reward model for better alignment [52, 57] with human preference using RL algorithms such as Proximal Policy Optimization (PPO; [51]). Meanwhile, PPO-style algorithm is also known for its instability, sample-inefficiency, and especially, a high demand for proper hyperparameter tuning [22]. This thus casts prohibitive computational cost to make the most effectiveness of PPO-based RLHF methods to align LLMs, especially for the open-source community.

Given that, further research on RLHF has explored various alternatives to PPO-based methods, with the most popular approach being the direct preference optimization method [82, 46], which skips the reward learning phase and directly optimizes the LLM to align it with the human preference. Our practical implementation (RPO) also harnesses the wisdom of reward-LLM equivalence to avoid explicit reward learning followed by PPO training.

Besides the original DPO algorithm [46], ever since it popularizing the direct preference learning style method, variants of the direct preference learning approach are proposed, including but not limited to [34, 5, 71, 59, 28, 74, 44, 26, 50, 32, 80, 58, 68, 30]. Each of them aims to address further challenges of direct preference learning from varying perspectives. Specifically, the algorithm proposed by [44, 26] share similar algorithmic components as RPO proposed in this work. Both work consider SFT style regularization during preference optimization. However, theoretical understanding of how SFT loss can help alignment remains unknown. In contrast, we provide theoretical justifications to the SFT loss as an implicit adversarial regularizer that provably mitigates overoptimization in preference learning.

**RLHF: theoretical investigation.**  Initiated from the literature of dueling bandits and dueling RL [76, 7, 43], recent success of RLHF in fine-tuning LLMs also motivates a long line of research to investigate the theoretical foundations of RLHF under different settings [11, 85, 78, 79, 67, 31, 71, 74, 19, 84], aiming to propose provably sample-efficient algorithms to learn a human-reward-maximizing policy from human preference signals. Our theoretical study of RLHF falls into the paradigm of offline learning from a pre-collected preference dataset, and is mostly related to the work of [85, 78, 31, 71, 74]. In this setup, the main challenge is to address the overoptimization issues due to human reward uncertainty and distributional shifts when only a fixed dataset is available. In the sequel, we compare our work with them in more detail.

Existing theoretical work on provably sample-efficient offline RLHF typically suffers from two drawbacks: they are either restricted to the linear function approximations setting [85, 71] which is far from the practical situations, or are generally unable to be implemented in the LLM experiments. Typically, to encompass the pessimistic principle in the face of uncertainty, the existing literature proposes to return the optimal policy against either an estimated reward model plus a structure-aware reward uncertainty penalty [71] or the most pessimistic reward model inside a confidence region [85, 78]. Both of these two types of method involve intractable components for implementation and needs for additional algorithmic design to approximate the theoretical algorithm in practice. In contrast, our theory works in the context of general function approximations while being friendly to be implemented. Finally, we remark that, while our study focuses on the standard Bradley-Terry model of human preference with general reward function approximations, the work of [74] further considers a general human preference model. But it remains unknown how their algorithms can be efficiently implemented in practice. It serves as an interesting direction to extend our technique to RLHF with general reward model and device new practical algorithms.

Finally, we mention that the algorithm design of RPO is also related to the "pessimism" principle in the standard offline RL literature. It proposes to maintain a pessimistic estimate of the policy values or constrain the policy not to take unseen actions in the data to handle the challenge of the insufficient coverage of the dataset, e.g., [29, 65, 69, 70, 47, 75, 72, 36, 54, 77, 39, 48, 53, 8, 38, 33, 24]. In contrast, we consider the offline RLHF problem and the techniques to obtain the objective of the

RPO algorithm (see Section 4) along with its sample complexity analysis are new and different from these works.

**Reward hacking and overoptimization in RLHF for LLM.** As is discussed in the introduction, the challenge of reward hacking or overoptimization may prevent the successful alignment of LLMs, degenerating the performance of an LLM because of maximizing an imperfect, overfitted, and misgeneralized proxy reward learned from the finite data [40, 62, 25, 10]. Efforts have been made to mitigate this fundamental issue through the perspective of theory, e.g., [85, 71, 86], and practice, e.g., [15, 21, 41, 81, 49, 56]. Our approach starts from the theoretical insights of handling inherent uncertainty in learning human preference from finite data, while being surprisingly easy to implement.

## B  Limitations and Future Works

One limitation of the current work is that we focus on the setting of offline RLHF where only a fixed preference dataset is available. Recent RLHF research has shown great potential of using iterative methods for LLM alignment with multiple rounds of preference data collection and tuning [71, 58].

Future works include extending our idea of theoretical algorithm design and analysis to the iterative RLHF setup where further preference data can be collected. Also, since our practical algorithm RPO is a plug-in module that effectively mitigates overoptimization and improves alignment performance, it serves as an exciting direction to combine it with explorative preference data collecting mechanism in iterative RLHF to further boost the performance of LLM alignment.

## C  Further Discussions

**Discussions on Algorithm 1 and Theorem 5.3.** We compare our theory with [71] and [78].

**Remark C.1** (Comparison with [71])**.** *Another theoretical work on RLHF [71] explicitly models the KL-regularization between the target policy and the reference policy in the learning objective, referred to as the KL-regularized contextual bandit. This means that their metric becomes the KL-regularized expected reward. In contrast, here we put the KL-regularization as a component of our algorithm design, but we still keep the metric as the expected reward (2.2). Therefore our theory in Section 5.1 directly reveals how the learned policy performs in terms of the expected reward compared to any given target policy (which can be a stochastic policy).*

**Remark C.2** (Comparison with [78])**.** *We remark that in the work of [78], they also mentioned a maximin object similar to (3.2) for offline preference-based RL as a complementary to their theoretical algorithm. However, the sample complexity of the maximin-style algorithm they presented is unknown, while we provide finite sample convergence result for Algorithm 1 in Section 5. Furthermore, our objective (3.2) features another KL-regularization term, which is essential for the proposal of our new practical algorithm design for aligning LLM in Section 4.*

**Discussions on the partial coverage assumption (Assumption 5.2).** A sufficient condition to make this partial coverage condition (Assumption 5.2) hold is that the distribution of the offline dataset, which is $\mu_{\mathcal{D}}$, can well cover the joint distribution of $(a^1, a^0) \sim (\pi, \pi^{\text{base}})$. Here to discuss focus on $\pi^{\text{base}} = \pi^{\text{chosen}}$ as we adopted in the experiment part.

First, we clarify that the offline dataset distribution $\mu_{\mathcal{D}}$ is not simply $(a^1, a^0) \sim (\pi^{\text{unchosen}}, \pi^{\text{chosen}})$, since according to our definition (see Section 2) whether $a^1$ or $a^0$ is chosen is random and is determined by $y \in 0, 1$ obeying the BT model. Thus, $(a^1, a^0) \sim \mu_{\mathcal{D}}$ can be interpreted as a mixture of $(\pi^{\text{unchosen}}, \pi^{\text{chosen}})$ and $(\pi^{\text{chosen}}, \pi^{\text{unchosen}})$. This mixture probability would not be too small as long as the quality of $(a^1, a^0)$ does not vary too much, i.e., both of them are possible to be chosen, which is the case in practice. As a result, in the offline data distribution $(a^1, a^0) \sim \mu_{\mathcal{D}}$, both $a^1$ and $a^0$ partly comes from the chosen distribution $\pi^{\text{chosen}}$.

Then in order for $\mu_{\mathcal{D}}$ to cover the joint distribution of $(a^1, a^0) \sim (\pi, \pi^{\text{base}})$, it suffices to argue that $\pi^{\text{chosen}}$ can cover the target policy $\pi$, which is then reduced back to the traditional coverage condition. Thus our assumption essentially requires that $\pi^{\text{chosen}}$ well covers and only needs to cover the target policy $\pi$. This coincides with the spirit of the minimal data assumption in offline RL theory, i.e., the so-called partial coverage condition.

**On the relationship between observed chosen probability and reward overoptimization.** First, we note that the actions and their chosen probabilities can be interpreted as a proxy of analyzing the underlying (estimated) reward model $\widehat{r}$ due to the representation $\pi_{\widehat{r}}(a|x) \propto \pi^{\text{ref}}(a|x) \exp(\beta^{-1}\widehat{r}(x,a))$. Analyzing the (log) probabilities of the actions can be utilized to detect the mitigation of overoptimization, because according to the representation, an overestimated reward of a poor action would result in a higher probability of choosing this action, and would also cause a decay in the probability of choosing other better actions (since the probabilities are normalized to 1).

To further showcase the ability of RPO to address overoptimization (through the lense of probability), consider the following theoretical example with only one state and three actions [73] where we can track everything clearly. It has three actions $a, b, c$ with $R^{\star}(a) = 1, R^{\star}(b) = 0.5, R^{\star}(c) = 0$. The reference policy $\pi^{\text{ref}}(a) = \pi^{\text{ref}}(b) = 0.4, \pi^{\text{ref}}(c) = 0.1$, and the dataset consists of one data point $\mathcal{D} = (a, b, 1)$ (meaning action $a$ is preferred in the data). Then an ideally solved DPO objective would be $\pi_{\text{DPO}}$ as long as $\pi^{\text{DPO}}(b) = 0$, and the value of $\pi^{\text{DPO}}(a)$ can be arbitrarily chosen in $[0, 1]$. Thus a possible solution to DPO would be $\pi^{\text{DPO}}(a) = 0.5, \pi^{\text{DPO}}(b) = 0$, and by the normalizing condition $\pi^{\text{DPO}}(c) = 0.5$, which is undesirable since the action $c$ has reward $R^{\star}(c) = 0$. In contrast, solving the RPO objective would additionally require the maximization of $\pi_{\text{RPO}}(a)$ due to the SFT regularization term, and thus the solution is shifted towards $\pi_{\text{RPO}}(a) = 1, \pi_{\text{RPO}}(b) = \pi_{\text{RPO}}(c) = 0$, which is better than the DPO policy. Thus, RPO is able to prevent overoptimization towards poor actions that are less covered by the dataset (action $c$ here), therefore resulting in a better policy.

**About the relationships and distinctions between PTX loss in [60] and the SFT loss of RPO.** The original PTX loss is an imitation loss calculated on the pretraining data. In contrast, the SFT loss in the RPO objective is an imitation loss calculated on the RLHF dataset. In more specific, our experiments use this SFT loss to imitate the chosen responses in the RLHF dataset. Thus the relationship is that they are both imitation loss which aims to mimic certain data distribution. The distinction is that they are calculated on different data sources. The SFT loss in the RPO objective naturally comes from our theoretical algorithm and provably serves as an important regularization term to mitigate overoptimization in offline RLHF.

**About the computational complexity of the SFT loss gradient.** According to the paragraph **Practical implementation** in Section 6, RPO adds an additional SFT loss (the log probability of the chosen labels in the preference dataset) on the original DPO loss, where we highlight that the SFT loss is actually an intermediate quantity in the calculation of the DPO loss. Hence, our proposed method does not incur any additional computation overhead compared with the vanilla DPO.

# D Proofs for Sample Complexity Analysis

## D.1 Proof of Theorem 5.3

*Proof of Theorem 5.3.* By definition, the suboptimality gap of $\widehat{\pi}$ w.r.t. $\pi$ is decomposed as following,

$$
\begin{aligned}
&\text{Gap}^{\pi}(\widehat{\pi}) \\
&= \mathbb{E}_{x\sim d_0, a\sim\pi(\cdot|x)}\big[r^{\star}(x,a)\big] - \mathbb{E}_{x\sim d_0, a\sim\widehat{\pi}(\cdot|x)}\big[r^{\star}(x,a)\big] \\
&= \mathbb{E}_{x\sim d_0, a^1\sim\pi(\cdot|x), a^0\sim\pi^{\text{ref}}(\cdot|x)}\Big[r^{\star}(x,a^1) - r^{\star}(x,a^0) - \beta\cdot\text{KL}\big(\pi(\cdot|x)\|\pi^{\text{ref}}(\cdot|x)\big)\Big] \\
&\quad - \eta^{-1}\cdot\min_{r\in\mathcal{R}}\left\{\eta\cdot\mathbb{E}_{\substack{x\sim d_0, a^1\sim\widehat{\pi}(\cdot|x),\\ a^0\sim\pi^{\text{base}}(\cdot|x)}}\Big[r(x,a^1) - r(x,a^0) - \beta\cdot\text{KL}\big(\widehat{\pi}(\cdot|x)\|\pi^{\text{ref}}(\cdot|x)\big)\Big] + \mathcal{L}_{\mathcal{D}}(r)\right\} \\
&\quad + \eta^{-1}\cdot\min_{r\in\mathcal{R}}\left\{\eta\cdot\mathbb{E}_{\substack{x\sim d_0, a^1\sim\widehat{\pi}(\cdot|x),\\ a^0\sim\pi^{\text{base}}(\cdot|x)}}\Big[r(x,a^1) - r(x,a^0) - \beta\cdot\text{KL}\big(\widehat{\pi}(\cdot|x)\|\pi^{\text{ref}}(\cdot|x)\big)\Big] + \mathcal{L}_{\mathcal{D}}(r)\right\} \\
&\quad - \mathbb{E}_{x\sim d_0, a^1\sim\widehat{\pi}(\cdot|x), a^0\sim\pi^{\text{base}}(\cdot|x)}\Big[r^{\star}(x,a^1) - r^{\star}(x,a^0) - \beta\cdot\text{KL}\big(\widehat{\pi}(\cdot|x)\|\pi^{\text{ref}}(\cdot|x)\big)\Big] \\
&\quad + \beta\cdot\mathbb{E}_{x\sim d_0}\Big[\text{KL}\big(\pi(\cdot|x)\|\pi^{\text{ref}}(\cdot|x)\big) - \text{KL}\big(\widehat{\pi}(\cdot|x)\|\pi^{\text{ref}}(\cdot|x)\big)\Big] \\
&:= \text{Term (A)} + \text{Term (B)} + \text{Term (C)}, \qquad\qquad\qquad\qquad\qquad\qquad\qquad (\text{D.1})
\end{aligned}
$$

where in the above Term (A), Term (B), and Term (C) are abbreviations for

Term (A)

$$
\begin{aligned}
&= \mathbb{E}_{x\sim d_0, a^1\sim\pi(\cdot|x), a^0\sim\pi^{\text{base}}(\cdot|x)}\Big[r^{\star}(x,a^1) - r^{\star}(x,a^0) - \beta\cdot\text{KL}\big(\pi(\cdot|x)\|\pi^{\text{ref}}(\cdot|x)\big)\Big] \\
&\quad - \eta^{-1}\cdot\min_{r\in\mathcal{R}}\left\{\eta\cdot\mathbb{E}_{\substack{x\sim d_0, a^1\sim\widehat{\pi}(\cdot|x),\\ a^0\sim\pi^{\text{base}}(\cdot|x)}}\Big[r(x,a^1) - r(x,a^0) - \beta\cdot\text{KL}\big(\widehat{\pi}(\cdot|x)\|\pi^{\text{ref}}(\cdot|x)\big)\Big] + \mathcal{L}_{\mathcal{D}}(r)\right\},
\end{aligned}
$$

and

Term (B)

$$
\begin{aligned}
&= \eta^{-1}\cdot\min_{r\in\mathcal{R}}\left\{\eta\cdot\mathbb{E}_{\substack{x\sim d_0, a^1\sim\widehat{\pi}(\cdot|x),\\ a^0\sim\pi^{\text{base}}(\cdot|x)}}\Big[r(x,a^1) - r(x,a^0) - \beta\cdot\text{KL}\big(\widehat{\pi}(\cdot|x)\|\pi^{\text{ref}}(\cdot|x)\big)\Big] + \mathcal{L}_{\mathcal{D}}(r)\right\} \\
&\quad - \mathbb{E}_{x\sim d_0, a^1\sim\widehat{\pi}(\cdot|x), a^0\sim\pi^{\text{base}(\cdot|x)}}\Big[r^{\star}(x,a^1) - r^{\star}(x,a^0) - \beta\cdot\text{KL}\big(\widehat{\pi}(\cdot|x)\|\pi^{\text{ref}}(\cdot|x)\big)\Big],
\end{aligned}
$$

and

$$
\text{Term (C)} = \beta\cdot\mathbb{E}_{x\sim d_0}\Big[\text{KL}\big(\pi(\cdot|x)\|\pi^{\text{ref}}(\cdot|x)\big) - \text{KL}\big(\widehat{\pi}(\cdot|x)\|\pi^{\text{ref}}(\cdot|x)\big)\Big].
$$

In the following, we analyze Term (A) and Term (B) respectively.

**Upper bound Term (A).** Notice that by the optimality of our choice of policy $\widehat{\pi}$ in (3.2), we have

Term (A)

$$= \mathbb{E}_{x \sim d_0, a^1 \sim \pi(\cdot|x), a^0 \sim \pi^{\text{base}}(\cdot|x)} \Big[ r^\star(x, a^1) - r^\star(x, a^0) - \beta \cdot \text{KL}\big(\pi(\cdot|x) \| \pi^{\text{ref}}(\cdot|x)\big) \Big] \quad \text{(D.2)}$$

$$- \eta^{-1} \cdot \min_{r \in \mathcal{R}} \left\{ \eta \cdot \mathbb{E}_{\substack{x \sim d_0, a^1 \sim \widehat{\pi}(\cdot|x), \\ a^0 \sim \pi^{\text{base}}(\cdot|x)}} \Big[ r(x, a^1) - r(x, a^0) - \beta \cdot \text{KL}\big(\widehat{\pi}(\cdot|x) \| \pi^{\text{ref}}(\cdot|x)\big) \Big] + \mathcal{L}_{\mathcal{D}}(r) \right\}$$

$$\leq \mathbb{E}_{x \sim d_0, a^1 \sim \pi(\cdot|x), a^0 \sim \pi^{\text{ref}}(\cdot|x)} \Big[ r^\star(x, a^1) - r^\star(x, a^0) - \beta \cdot \text{KL}\big(\pi(\cdot|x) \| \pi^{\text{ref}}(\cdot|x)\big) \Big]$$

$$- \eta^{-1} \cdot \min_{r \in \mathcal{R}} \left\{ \eta \cdot \mathbb{E}_{\substack{x \sim d_0, a^1 \sim \pi(\cdot|x), \\ a^0 \sim \pi^{\text{base}}(\cdot|x)}} \Big[ r(x, a^1) - r(x, a^0) - \beta \cdot \text{KL}\big(\pi(\cdot|x) \| \pi^{\text{ref}}(\cdot|x)\big) \Big] + \mathcal{L}_{\mathcal{D}}(r) \right\}$$

$$= \max_{r \in \mathcal{R}} \left\{ \mathbb{E}_{x \sim d_0, a^1 \sim \pi(\cdot|x), a^0 \sim \pi^{\text{base}}(\cdot|x)} \Big[ \big(r^\star(x, a^1) - r^\star(x, a^0)\big) - \big(r(x, a^1) - r(x, a^0)\big) \Big] - \eta^{-1} \cdot \mathcal{L}_{\mathcal{D}}(r) \right\},$$

where in the inequality we apply the optimality of the choice of policy $\widehat{\pi}$ in (3.2).

**Upper bound Term (B).** For this term, we directly consider the following bound,

Term (B)

$$= \eta^{-1} \cdot \min_{r \in \mathcal{R}} \left\{ \eta \cdot \mathbb{E}_{\substack{x \sim d_0, a^1 \sim \widehat{\pi}(\cdot|x), \\ a^0 \sim \pi^{\text{ref}}(\cdot|x)}} \Big[ r(x, a^1) - r(x, a^0) - \beta \cdot \text{KL}\big(\widehat{\pi}(\cdot|x) \| \pi^{\text{ref}}(\cdot|x)\big) \Big] + \mathcal{L}_{\mathcal{D}}(r) \right\}$$

$$- \mathbb{E}_{x \sim d_0, a^1 \sim \widehat{\pi}(\cdot|x), a^0 \sim \pi^{\text{base}}(\cdot|x)} \Big[ r^\star(x, a^1) - r^\star(x, a^0) - \beta \cdot \text{KL}\big(\widehat{\pi}(\cdot|x) \| \pi^{\text{ref}}(\cdot|x)\big) \Big]$$

$$\leq \mathbb{E}_{x \sim d_0, a^1 \sim \widehat{\pi}(\cdot|x), a^0 \sim \pi^{\text{base}}(\cdot|x)} \Big[ r^\star(x, a^1) - r^\star(x, a^0) - \beta \cdot \text{KL}\big(\widehat{\pi}(\cdot|x) \| \pi^{\text{ref}}(\cdot|x)\big) \Big] + \eta^{-1} \cdot \mathcal{L}_{\mathcal{D}}(r^\star)$$

$$- \mathbb{E}_{x \sim d_0, a^1 \sim \widehat{\pi}(\cdot|x), a^0 \sim \pi^{\text{base}}(\cdot|x)} \Big[ r^\star(x, a^1) - r^\star(x, a^0) - \beta \cdot \text{KL}\big(\widehat{\pi}(\cdot|x) \| \pi^{\text{ref}}(\cdot|x)\big) \Big]$$

$$= \eta^{-1} \cdot \mathcal{L}_{\mathcal{D}}(r^\star), \quad \text{(D.3)}$$

where in the inequality we apply the fact that $r^\star \in \mathcal{R}$ by Assumption 5.1.

**Combining Term (A), Term (B), and Term (C).** Now by (D.1), (D.2), and (D.3), we have that

$$\text{Gap}_\beta^\pi(\widehat{\pi}) = \text{Term (A)} + \text{Term (B)} + \text{Term (C)} \quad \text{(D.4)}$$

$$\leq \max_{r \in \mathcal{R}} \left\{ \mathbb{E}_{\substack{x \sim d_0, a^1 \sim \pi(\cdot|x), \\ a^0 \sim \pi^{\text{base}}(\cdot|x)}} \Big[ \big(r^\star(x, a^1) - r^\star(x, a^0)\big) - \big(r(x, a^1) - r(x, a^0)\big) \Big] + \eta^{-1} \cdot \big(\mathcal{L}_{\mathcal{D}}(r^\star) - \mathcal{L}_{\mathcal{D}}(r)\big) \right\}$$

$$+ \beta \cdot \mathbb{E}_{x \sim d_0} \Big[ \text{KL}\big(\pi(\cdot|x) \| \pi^{\text{ref}}(\cdot|x)\big) - \text{KL}\big(\widehat{\pi}(\cdot|x) \| \pi^{\text{ref}}(\cdot|x)\big) \Big].$$

In the following, we upper bound the right hand side of (D.4) via relating the MLE loss difference term to the reward difference term through a careful analysis of the preference model. On the one hand, we invoke Lemma D.1 to give an upper bound of the difference of the MLE loss as following, with probability at least $1 - \delta$ over random samples and $\varepsilon = (6 \cdot (1 + e^R) \cdot N)^{-1}$, for any reward model $r \in \mathcal{R}$, it holds that

$$\mathcal{L}_{\mathcal{D}}(r^\star) - \mathcal{L}_{\mathcal{D}}(r)$$

$$\leq -2 \cdot \mathbb{E}_{(x, a^1, a^0) \sim \mu_{\mathcal{D}}(\cdot, \cdot, \cdot)} \Big[ D_{\text{Hellinger}}^2 \big(\mathbb{P}_{r^\star}(\cdot|x, a^1, a^0) \| \mathbb{P}_r(\cdot|x, a^1, a^0)\big) \Big]$$

$$+ \frac{3}{N} \cdot \log\left(\frac{\mathcal{N}_\varepsilon(\mathcal{R}, \|\cdot\|_\infty)}{\delta}\right),$$

where we recall that we use the subscript $r$ in $\mathbb{P}_r$ to emphasize the dependence of the probabilistic model on the reward model. Here $\mathcal{N}_\varepsilon(\mathcal{R}, \|\cdot\|_\infty)$ denotes the $\varepsilon$-covering number of the reward model class and $R$ is the upper bound on the reward functionss (Assumption 5.1). Now to facilitate the calculation, we lower bound the Hellinger distance by total variation (TV) distance as

$$D_{\text{Hellinger}}^2\big(\mathbb{P}_{r^\star}(\cdot|x, a^1, a^0)\|\mathbb{P}_r(\cdot|x, a^1, a^0)\big) \geq D_{\text{TV}}^2\big(\mathbb{P}_{r^\star}(\cdot|x, a^1, a^0)\|\mathbb{P}_r(\cdot|x, a^1, a^0)\big),$$

By the expression of the probability model $\mathbb{P}_r$, we can further write the TV distance above as

$$
\begin{aligned}
D_{\text{TV}}&\big(\mathbb{P}_{r^\star}(\cdot|x, a^1, a^0)\|\mathbb{P}_r(\cdot|x, a^1, a^0)\big) \\
&= \frac{1}{2} \cdot \Big|\sigma\big(r^\star(x, a^1) - r^\star(x, a^0)\big) - \sigma\big(r(x, a^1) - r(x, a^0)\big)\Big| \\
&\quad + \frac{1}{2} \cdot \Big|\sigma\big(r^\star(x, a^0) - r^\star(x, a^1)\big) - \sigma\big(r(x, a^0) - r(x, a^1)\big)\Big| \\
&= \Big|\sigma\big(r^\star(x, a^1) - r^\star(x, a^0)\big) - \sigma\big(r(x, a^1) - r(x, a^0)\big)\Big|,
\end{aligned}
\tag{D.5}
$$

where in the second equality we use the fact that $\sigma(-z) = 1 - \sigma(z)$. Now by Lemma D.2 and the condition that $r(x, a) \in [0, R]$ for any $(x, a, r) \in \mathcal{X} \times \mathcal{A} \times \mathcal{R}$ (Assumption 5.1), we know that

$$
\begin{aligned}
\Big|\sigma\big(r^\star(x, a^1) - r^\star(x, a^0)\big) &- \sigma\big(r(x, a^1) - r(x, a^0)\big)\Big| \\
&\geq \kappa \cdot \Big|\big(r^\star(x, a^1) - r^\star(x, a^0)\big) - \big(r(x, a^1) - r(x, a^0)\big)\Big|,
\end{aligned}
$$

where $\kappa = 1/(1 + \exp(R))^2$. As a result, the difference of the MLE loss is upper bounded by

$$
\begin{aligned}
\mathcal{L}_\mathcal{D}(r^\star) &- \mathcal{L}_\mathcal{D}(r) \\
&\leq -2\kappa^2 \cdot \mathbb{E}_{(x, a^1, a^0) \sim \mu_\mathcal{D}(\cdot, \cdot, \cdot)}\left[\Big|\big(r^\star(x, a^1) - r^\star(x, a^0)\big) - \big(r(x, a^1) - r(x, a^0)\big)\Big|^2\right] \\
&\quad + \frac{3}{N} \cdot \log\left(\frac{\mathcal{N}_\varepsilon(\mathcal{R}, \|\cdot\|_\infty)}{\delta}\right).
\end{aligned}
\tag{D.6}
$$

On the other hand, the reward difference term in (D.4), which is evaluated on actions from $\pi$ and $\pi^{\text{base}}$, can be related to the reward difference evaluated on the data distribution $\mu_\mathcal{D}$ via Assumption 5.2, i.e.,

$$\mathbb{E}_{x \sim d_0, a^1 \sim \pi(\cdot|x), a^0 \sim \pi^{\text{base}}(\cdot|x)}\left[\big(r^\star(x, a^1) - r^\star(x, a^0)\big) - \big(r(x, a^1) - r(x, a^0)\big)\right] \tag{D.7}$$

$$\leq C_{\mu_\mathcal{D}}(\mathcal{R}; \pi, \pi^{\text{base}})\sqrt{\mathbb{E}_{(x, a^1, a^0) \sim \mu_\mathcal{D}}\left[\Big|\big(r^\star(x, a^1) - r^\star(x, a^0)\big) - \big(r(x, a^1) - r(x, a^0)\big)\Big|^2\right]}.$$

Finally, combining (D.6), (D.7), and (D.4), denoting

$$\Delta_r := \sqrt{\mathbb{E}_{(x, a^1, a^0) \sim \mu_\mathcal{D}}\left[\Big|\big(r^\star(x, a^1) - r^\star(x, a^0)\big) - \big(r(x, a^1) - r(x, a^0)\big)\Big|^2\right]},$$

we have that

$$
\begin{aligned}
\text{Gap}^\pi(\widehat{\pi}) &\leq \max_{r \in \mathcal{R}}\left\{C_{\mu_\mathcal{D}}(\mathcal{R}; \pi, \pi^{\text{base}}) \cdot \Delta_r - 2\eta^{-1}\kappa^2 \cdot \Delta_r^2\right\} + \frac{3}{\eta N} \cdot \log\left(\frac{\mathcal{N}_\varepsilon(\mathcal{R}, \|\cdot\|_\infty)}{\delta}\right) \\
&\quad + \beta \cdot \mathbb{E}_{x \sim d_0}\left[\text{KL}\big(\pi(\cdot|x)\|\pi^{\text{ref}}(\cdot|x)\big) - \text{KL}\big(\widehat{\pi}(\cdot|x)\|\pi^{\text{ref}}(\cdot|x)\big)\right] \\
&\leq \frac{\big(C_{\mu_\mathcal{D}}(\mathcal{R}; \pi, \pi^{\text{base}})\big)^2 \eta}{8\kappa^2} + \frac{3}{\eta N} \cdot \log\left(\frac{\mathcal{N}_\varepsilon(\mathcal{R}, \|\cdot\|_\infty)}{\delta}\right) \\
&\quad + \beta \cdot \mathbb{E}_{x \sim d_0}\left[\text{KL}\big(\pi(\cdot|x)\|\pi^{\text{ref}}(\cdot|x)\big)\right],
\end{aligned}
$$

where in the second inequality we use that fact that $az - bz^2 \leq a^2/(4b)$ for any $z \in \mathbb{R}$ and that KL-divergence is non-negative. Consequently, with the choice of

$$\eta = 2\sqrt{6} \cdot \sqrt{\frac{\log\left(\mathcal{N}_\varepsilon(\mathcal{R}, \|\cdot\|_\infty)/\delta\right)}{N}}, \quad \beta = \frac{1}{\sqrt{N}}, \quad \kappa = \frac{1}{(1 + \exp(R))^2},$$

we conclude that with probability at least $1 - \delta$ and $\varepsilon = (6 \cdot (1 + e^R) \cdot N)^{-1}$,

$\mathrm{Gap}^\pi(\widehat{\pi})$

$$\leq \frac{\sqrt{6}\left(1 + \exp(R)\right)^2 \left(\left(C_{\mu_\mathcal{D}}(\mathcal{R}; \pi, \pi^{\mathrm{base}})\right)^2 + 1\right) \iota + 4\mathbb{E}_{x \sim d_0}\left[\mathrm{KL}\left(\pi(\cdot|x)\|\pi^{\mathrm{ref}}(\cdot|x)\right)\right]}{4\sqrt{N}},$$

where we denote $\iota = \sqrt{\log\left(\mathcal{N}_\varepsilon(\mathcal{R}, \|\cdot\|_\infty)/\delta\right)}$ with $\varepsilon = (6 \cdot (1 + e^R) \cdot N)^{-1}$. This finishes the proof of Theorem 5.3. $\qquad\square$

## D.2 Technical Lemmas

**Lemma D.1** (Uniform concentration). *Consider the MLE loss* (3.1) *and define the approximation error as* $\varepsilon = (6 \cdot (1 + e^R) \cdot N)^{-1}$ *where* $R$ *is the upper bound on the reward functions (Assumption 5.2). Suppose that the reward model class* $\mathcal{R}$ *has a finite* $\varepsilon$-*covering number* $\mathcal{N}_\varepsilon(\mathcal{R}, \|\cdot\|_\infty) < \infty$. *Then for any* $\delta < 1/e$ *it holds with probability at least* $1 - \delta$ *that*

$$\mathcal{L}_\mathcal{D}(r^\star) - \mathcal{L}_\mathcal{D}(r)$$
$$\leq -2 \cdot \mathbb{E}_{(x,a^1,a^0) \sim \mu_\mathcal{D}(\cdot,\cdot,\cdot)}\left[D^2_{\mathrm{Hellinger}}\left(\mathbb{P}_{r^\star}(\cdot|x,a^1,a^0)\|\mathbb{P}_r(\cdot|x,a^1,a^0)\right)\right]$$
$$+ \frac{3}{N} \cdot \log\left(\frac{\mathcal{N}_\varepsilon(\mathcal{R}, \|\cdot\|_\infty)}{\delta}\right).$$

*Proof of Lemma D.1.* For notational simplicity, we use $\mathcal{C}_\varepsilon(\mathcal{R}, \|\cdot\|_\infty)$ to denote an $\varepsilon$-cover of the reward model class $\mathcal{R}$ under the $\|\cdot\|_\infty$-norm. It holds that $\mathcal{N}_\varepsilon(\mathcal{R}, \|\cdot\|_\infty) = |\mathcal{C}_\varepsilon(\mathcal{R}, \|\cdot\|_\infty)|$. First we invoke Proposition 5.3 of [37] to obtain a uniform concentration over the finite set of $\varepsilon$-cover $\mathcal{C}_\varepsilon(\mathcal{R}, \|\cdot\|_\infty)$. Specifically, with probability at least $1 - \delta$, for any $r \in \mathcal{C}_\varepsilon(\mathcal{R}, \|\cdot\|_\infty)$,

$$\mathcal{L}_\mathcal{D}(r^\star) - \mathcal{L}_\mathcal{D}(r)$$
$$\leq -2 \cdot \mathbb{E}_{(x,a^1,a^0) \sim \mu_\mathcal{D}(\cdot,\cdot,\cdot)}\left[D^2_{\mathrm{Hellinger}}\left(\mathbb{P}_{r^\star}(\cdot|x,a^1,a^0)\|\mathbb{P}_r(\cdot|x,a^1,a^0)\right)\right]$$
$$+ \frac{2}{N} \cdot \log\left(\frac{\mathcal{N}_\varepsilon(\mathcal{R}, \|\cdot\|_\infty)}{\delta}\right). \tag{D.8}$$

Now for any reward model $r \in \mathcal{R}$, we take a $r^\dagger \in \mathcal{C}_\varepsilon(\mathcal{R}, \|\cdot\|_\infty)$ satisfying $\|r - r^\dagger\|_\infty \leq \varepsilon$. We have

$$\mathcal{L}_\mathcal{D}(r^\star) - \mathcal{L}_\mathcal{D}(r)$$
$$= \mathcal{L}_\mathcal{D}(r^\star) - \mathcal{L}_\mathcal{D}(r^\dagger) + \mathcal{L}_\mathcal{D}(r^\dagger) - \mathcal{L}_\mathcal{D}(r)$$
$$\leq -2 \cdot \mathbb{E}_{(x,a^1,a^0) \sim \mu_\mathcal{D}(\cdot,\cdot,\cdot)}\left[D^2_{\mathrm{Hellinger}}\left(\mathbb{P}_{r^\star}(\cdot|x,a^1,a^0)\|\mathbb{P}_{r^\dagger}(\cdot|x,a^1,a^0)\right)\right]$$
$$\quad + \frac{2}{N} \cdot \log\left(\frac{\mathcal{N}_\varepsilon(\mathcal{R}, \|\cdot\|_\infty)}{\delta}\right) + \mathcal{L}_\mathcal{D}(r^\dagger) - \mathcal{L}_\mathcal{D}(r)$$
$$\leq -2 \cdot \mathbb{E}_{(x,a^1,a^0) \sim \mu_\mathcal{D}(\cdot,\cdot,\cdot)}\left[D^2_{\mathrm{Hellinger}}\left(\mathbb{P}_{r^\star}(\cdot|x,a^1,a^0)\|\mathbb{P}_r(\cdot|x,a^1,a^0)\right)\right]$$
$$\quad + \frac{2}{N} \cdot \log\left(\frac{\mathcal{N}_\varepsilon(\mathcal{R}, \|\cdot\|_\infty)}{\delta}\right) + \mathcal{L}_\mathcal{D}(r^\dagger) - \mathcal{L}_\mathcal{D}(r)$$
$$\quad + 4 \cdot \mathbb{E}_{(x,a^1,a^0) \sim \mu_\mathcal{D}(\cdot,\cdot,\cdot)}\left[D^2_{\mathrm{Hellinger}}\left(\mathbb{P}_{r^\dagger}(\cdot|x,a^1,a^0)\|\mathbb{P}_r(\cdot|x,a^1,a^0)\right)\right], \tag{D.9}$$

where in the fir inequality we use (D.8) for $r^\dagger$ and in the second inequality we utilize the triangular inequality for Hellinger distance. Therefore, it remains to upper bound the approximation error induced by $r^\dagger$. On the one hand, by the definition of $\mathcal{L}_\mathcal{D}$ in (3.1), we have that

$$\mathcal{L}_\mathcal{D}(r^\dagger) - \mathcal{L}_\mathcal{D}(r)$$

$$= \frac{1}{N} \sum_{i=1}^{N} y_i \cdot \log \left( \frac{\sigma\big(r(x_i, a_i^1) - r(x_i, a_i^0)\big)}{\sigma\big(r^\dagger(x_i, a_i^1) - r^\dagger(x_i, a_i^0)\big)} \right)$$

$$+ \frac{1}{N} \sum_{i=1}^{N} (1 - y_i) \cdot \log \left( \frac{\sigma\big(r(x_i, a_i^0) - r(x_i, a_i^1)\big)}{\sigma\big(r^\dagger(x_i, a_i^0) - r^\dagger(x_i, a_i^1)\big)} \right).$$

Use the inequality that $\log(x) \le x - 1$, we can further upper bound $\mathcal{L}_\mathcal{D}(r^\dagger) - \mathcal{L}_\mathcal{D}(r)$ by

$$\mathcal{L}_\mathcal{D}(r^\dagger) - \mathcal{L}_\mathcal{D}(r)$$

$$\le \frac{1}{N} \sum_{i=1}^{N} y_i \cdot \frac{\sigma\big(r(x_i, a_i^1) - r(x_i, a_i^0)\big) - \sigma\big(r^\dagger(x_i, a_i^1) - r^\dagger(x_i, a_i^0)\big)}{\sigma\big(r^\dagger(x_i, a_i^1) - r^\dagger(x_i, a_i^0)\big)}$$

$$+ \frac{1}{N} \sum_{i=1}^{N} (1 - y_i) \cdot \frac{\sigma\big(r(x_i, a_i^0) - r(x_i, a_i^1)\big) - \sigma\big(r^\dagger(x_i, a_i^0) - r^\dagger(x_i, a_i^1)\big)}{\sigma\big(r^\dagger(x_i, a_i^0) - r^\dagger(x_i, a_i^1)\big)}.$$

Now since $\|r^\dagger - r\|_\infty \le \varepsilon$ and $r^\dagger \in [0, R]$, invoking Lemma D.2, we can derive that

$$\mathcal{L}_\mathcal{D}(r^\dagger) - \mathcal{L}_\mathcal{D}(r) \le \frac{1}{N} \sum_{i=1}^{N} \frac{\left| \big(r(x_i, a_i^1) - r(x_i, a_i^0)\big) - \big(r^\dagger(x_i, a_i^1) - r^\dagger(x_i, a_i^0)\big) \right|}{(1 + e^R)^{-1}}$$

$$+ \frac{1}{N} \sum_{i=1}^{N} \frac{\left| \big(r(x_i, a_i^0) - r(x_i, a_i^1)\big) - \big(r^\dagger(x_i, a_i^0) - r^\dagger(x_i, a_i^1)\big) \right|}{(1 + e^R)^{-1}}$$

$$\le 4 \cdot \|r^\dagger - r\|_\infty \cdot (1 + e^R) \le 4\varepsilon \cdot (1 + e^R). \tag{D.10}$$

On the other hand, we upper bound the hellinger distance between $\mathbb{P}_r$ and $\mathbb{P}_{r^\dagger}$, for any $(x, a^1, a^0) \in \mathcal{X} \times \mathcal{A} \times \mathcal{A}$,

$$D^2_{\mathrm{Hellinger}}\big(\mathbb{P}_{r^\dagger}(\cdot | x, a^1, a^0) \| \mathbb{P}_r(\cdot | x, a^1, a^0)\big)$$

$$\le D_{\mathrm{TV}}\big(\mathbb{P}_{r^\dagger}(\cdot | x, a^1, a^0) \| \mathbb{P}_r(\cdot | x, a^1, a^0)\big)$$

$$= \left| \sigma\big(r^\dagger(x, a^1) - r^\dagger(x, a^0)\big) - \sigma\big(r(x, a^1) - r(x, a^0)\big) \right|$$

$$\le \left| \big(r^\dagger(x, a^1) - r^\dagger(x, a^0)\big) - \big(r(x, a^1) - r(x, a^0)\big) \right|$$

$$\le 2 \cdot \|r^\dagger - r\|_\infty \le 2\varepsilon, \tag{D.11}$$

where the first inequality uses the fact that $D^2_{\mathrm{Hellinger}} \le D_{\mathrm{TV}}$, the equality uses the same argument as (D.5), and the second inequality applies Lemma D.2. Finally, combining (D.9), (D.10), and (D.11), we conclude that

$$\mathcal{L}_\mathcal{D}(r^\star) - \mathcal{L}_\mathcal{D}(r) \le -2 \cdot \mathbb{E}_{(x, a^1, a^0) \sim \mu_\mathcal{D}(\cdot, \cdot, \cdot)} \left[ D^2_{\mathrm{Hellinger}}\big(\mathbb{P}_{r^\star}(\cdot | x, a^1, a^0) \| \mathbb{P}_r(\cdot | x, a^1, a^0)\big) \right]$$

$$+ \frac{2}{N} \cdot \log \left( \frac{\mathcal{N}_\varepsilon(\mathcal{R}, \| \cdot \|_\infty)}{\delta} \right) + 6\varepsilon \cdot (1 + e^R).$$

By taking the approximation error $\varepsilon = (6 \cdot (1 + e^R) \cdot N)^{-1}$, we conclude that for $\delta < e^{-1}$, with probability at least $1 - \delta$, for any $r \in \mathcal{R}$, it holds that

$$\mathcal{L}_\mathcal{D}(r^\star) - \mathcal{L}_\mathcal{D}(r)$$

$$\le -2 \cdot \mathbb{E}_{(x, a^1, a^0) \sim \mu_\mathcal{D}(\cdot, \cdot, \cdot)} \left[ D^2_{\mathrm{Hellinger}}\big(\mathbb{P}_{r^\star}(\cdot | x, a^1, a^0) \| \mathbb{P}_r(\cdot | x, a^1, a^0)\big) \right]$$

$$+ \frac{3}{N} \cdot \log \left( \frac{\mathcal{N}_\varepsilon(\mathcal{R}, \| \cdot \|_\infty)}{\delta} \right).$$

This completes the proof of Lemma D.1. $\qquad \square$

**Lemma D.2** (Sigmoid function). *For any real numbers $z_1, z_2 \in [-R, R]$, it holds that*

$$\kappa \cdot |z_1 - z_2| \le |\sigma(z_1) - \sigma(z_2)| \le |z_1 - z_2|,$$

*where the constant $\kappa = 1/(1 + \exp(R))^2$.*

*Proof of Lemma D.2.* Since the sigmoid function $\sigma(\cdot)$ is differentiable, we know that for any $z_1, z_2 \in [-R, R]$, there exists some $\xi(z_1, z_2) \in [-R, R]$ such that

$$\sigma(z_1) - \sigma(z_2) = \sigma'\big(\xi(z_1, z_2)\big) \cdot (z_1 - z_2).$$

Notice that $\sigma'(z) = \sigma(z) \cdot (1 - \sigma(z))$, we can obtain that

$$\begin{aligned}
1 \geq \sigma'\big(\xi(z_1, z_2)\big) &= \sigma\big(\xi(z_1, z_2)\big) \cdot \Big(1 - \sigma\big(\xi(z_1, z_2)\big)\Big) \\
&= \frac{1}{1 + \exp(\xi(z_1, z_2))} \cdot \left(1 - \frac{1}{1 + \exp(\xi(z_1, z_2))}\right) \\
&\geq \frac{1}{1 + \exp(R)} \cdot \left(1 - \frac{1}{1 + \exp(-R)}\right) \\
&= \frac{1}{(1 + \exp(R))^2}.
\end{aligned}$$

This completes the proof of Lemma D.2. $\qquad\square$

# E  Proofs for Equivalence between Maximin and Minimax Objectives

## E.1  Proof of Theorem 5.6

*Proof of Theorem 5.6.* Consider denoting an auxiliary policy $\widehat{\pi}$ as

$$\widehat{\pi} \in \underset{\pi \in \Pi}{\operatorname{argmax}} \ \min_{r \in \mathcal{R}} \phi(\pi, r). \tag{E.1}$$

By the definition of $\widehat{r}$ and $\widehat{\pi}$, the duality gap of $(\widehat{r}, \widehat{\pi})$, defined as

$$\operatorname{Dual}(\widehat{r}, \widehat{\pi}) := \max_{\pi \in \Pi} \phi(\pi, \widehat{r}) - \min_{r \in \mathcal{R}} \phi(\widehat{\pi}, r)$$

is zero. This is because the following deduction,

$$\begin{aligned}
\operatorname{Dual}(\widehat{r}, \widehat{\pi}) &= \left(\max_{\pi \in \Pi} \phi(\pi, \widehat{r}) - \min_{r \in \mathcal{R}} \max_{\pi \in \Pi} \phi(\pi, r)\right) \\
&\quad + \left(\max_{\pi \in \Pi} \min_{r \in \mathcal{R}} \phi(\pi, r) - \min_{r \in \mathcal{R}} \phi(\widehat{\pi}, r)\right) \\
&= 0, \tag{E.2}
\end{aligned}$$

where in the first equality we apply Lemma E.1 that the minimax objective and the maximin objective are equivalent, and the last equality applies the definition of $\widehat{r}$ and $\widehat{\pi}$ respectively. Note that we can rewrite the duality gap as following

$$\operatorname{Dual}(\widehat{r}, \widehat{\pi}) = \left(\max_{\pi \in \Pi} \phi(\pi, \widehat{r}) + \phi(\widehat{\pi}, \widehat{r})\right) - \left(\phi(\widehat{\pi}, \widehat{r}) - \min_{r \in \mathcal{R}} \phi(\widehat{\pi}, r)\right). \tag{E.3}$$

Combining (E.2) and (E.3), we can conclude that

$$\max_{\pi \in \Pi} \phi(\pi, \widehat{r}) = \phi(\widehat{\pi}, \widehat{r}) \quad \Rightarrow \quad \widehat{\pi} \in \underset{\pi \in \Pi}{\operatorname{argmax}} \ \phi(\widehat{r}, \pi). \tag{E.4}$$

Now comparing what $\pi_{\widehat{r}}$ and $\widehat{\pi}$ satisfy in (5.4) and (E.4) respectively, invoking Lemma E.3 that the maximizer of $\phi(\cdot, r)$ given any $r \in \mathcal{R}$ is unique on the support of $d_0$, we can conclude that

$$\pi_{\widehat{r}}(\cdot | x) = \widehat{\pi}(\cdot | x), \quad \forall x \in \operatorname{Supp}(d_0). \tag{E.5}$$

Therefore, by (E.1) and (E.5), and the fact that $\phi(\pi, r)$ depends on $\pi$ only through its value on the support of $d_0$, we can conclude that

$$\pi_{\widehat{r}} \in \underset{\pi \in \Pi}{\operatorname{argmax}} \ \min_{r \in \mathcal{R}} \phi(\pi, r).$$

This finishes the proof of Theorem 5.6. $\qquad\square$

### E.2 Auxiliary Lemmas

**Lemma E.1** (Equivalence of maximin and minimax objectives). *For the policy class $\Pi$ defined in (2.3) and the reward model class $\mathcal{R}$ satisfying Assumption 5.5, it holds that the maximin objective is equivalent to the minimax objective, i.e.,*

$$\max_{\pi \in \Pi} \min_{r \in \mathcal{R}} \phi(\pi, r) = \min_{r \in \mathcal{R}} \max_{\pi \in \Pi} \phi(\pi, r).$$

*Proof of Lemma E.1.* The foundation of this result is a minimax theorem given by [23] (Lemma E.2). In our setting, the policy class $\Pi$ is a nonempty set, and the reward model class $\mathcal{R}$ is a nonempty compact Hausdorff space. Furthermore, by our choice of the policy class $\Pi$ in (2.3), $\Pi$ is a convex set. Meanwhile, the function $\phi$ is a concave function of $\pi \in \Pi$ since the dependence on $\pi$ is linear terms plus a negative KL term (concave). Finally, by our assumption, the function $\phi$ is convex-like on the reward model class $\mathcal{R}$ and is also continuous on $\mathcal{R}$. As a result, all the conditions of Lemma E.2 are satisfied and the minimax theorem holds in our problem setup, finishing the proof of Lemma E.1. $\square$

**Lemma E.2** (Minimax theorem [23]). *Let $\mathcal{X}$ be a nonempty set (not necessarily topologized) and $\mathcal{Y}$ be a nonempty compact topological space. Let $f : \mathcal{X} \times \mathcal{Y} \mapsto \mathbb{R}$ be lower semicontinuous on $\mathcal{Y}$. Suppose that $f$ is concave-like on $\mathcal{X}$ and convex-like on $\mathcal{Y}$, i.e., for any $x_1, x_2 \in \mathcal{X}$, $\alpha \in [0, 1]$, there exists $x_3 \in \mathcal{X}$ such that*

$$f(x_3, \cdot) \geq \alpha \cdot f(x_1, \cdot) + (1 - \alpha) \cdot f(x_2, \cdot) \text{ on } \mathcal{Y},$$

*and for any $y_1, y_2 \in \mathcal{Y}$, $\beta \in [0, 1]$, there exists $y_3 \in \mathcal{Y}$ such that*

$$f(\cdot, y_3) \leq \beta \cdot f(\cdot, y_1) + (1 - \beta) \cdot f(\cdot, y_2) \text{ on } \mathcal{Y}.$$

*Then the following equation holds,*

$$\max_{x \in \mathcal{X}} \min_{y \in \mathcal{Y}} f(x, y) = \min_{y \in \mathcal{Y}} \max_{x \in \mathcal{X}} f(x, y).$$

**Lemma E.3** (Unique maximizer of $\phi$). *Consider the function $\phi$ defined as*

$$\phi(\pi, r) := \eta \cdot \mathbb{E}_{x \sim d_0, a^1 \sim \pi(\cdot|x), a^0 \sim \pi^{\mathrm{base}}(\cdot|x)} \Big[ r(x, a^1) - r(x, a^0) - \beta \cdot D_{\mathrm{KL}}\big(\pi(\cdot|x) \| \pi^{\mathrm{ref}}(\cdot|x)\big) \Big]$$
$$+ \mathcal{L}_{\mathcal{D}}(r).$$

*Then given any $r \in \mathcal{R}$, the maximimzer of $\phi(\cdot, r)$ is unique on the support of $d_0$.*

*Proof of Lemma E.3.* Given any $r \in \mathcal{R}$, consider that

$$\max_{\pi \in \Pi} \phi(\pi, r)$$
$$= \eta \cdot \max_{\pi \in \Pi} \Big\{ \mathbb{E}_{x \sim d_0, a^1 \sim \pi(\cdot|x)} \Big[ r(x, a^1) - \beta \cdot D_{\mathrm{KL}}\big(\pi(\cdot|x) \| \pi^{\mathrm{ref}}(\cdot|x)\big) \Big] \Big\}$$
$$= \eta \cdot \max_{\pi \in \Pi} \left\{ C_r - \beta \cdot \mathbb{E}_{x \sim d_0} \left[ D_{\mathrm{KL}} \left( \pi(\cdot|x) \middle\| \frac{\pi^{\mathrm{ref}}(\cdot|x) \cdot \exp(\beta^{-1} \cdot r(x, \cdot))}{\int_{a' \in \mathcal{A}} \mathrm{d}\pi^{\mathrm{ref}}(a'|x) \cdot \exp(\beta^{-1} \cdot r(x, a'))} \right) \right] \right\},$$

where

$$C_r = \mathbb{E}_{x \sim d_0} \left[ \beta \cdot \log \left( \int_{a \in \mathcal{A}} \mathrm{d}\pi^{\mathrm{ref}}(a|x) \cdot \exp\big(\beta^{-1} \cdot r(x, a)\big) \right) \right]$$

is a constant independent of $\pi$. Therefore, the maximizer of $\phi(\cdot, r)$ on the support of $d_0$ must equal to

$$\pi_r(\cdot|x) = \frac{\pi^{\mathrm{ref}}(\cdot|x) \cdot \exp(\beta^{-1} \cdot r(x, \cdot))}{\int_{a' \in \mathcal{A}} \mathrm{d}\pi^{\mathrm{ref}}(a'|x) \cdot \exp(\beta^{-1} \cdot r(x, a'))},$$

which completes the proof of Lemma E.3. $\square$

# F  Additional Details on Experiments

## F.1  Training Details

We train the gemma series models with 8 NVIDIA A6000 GPUs and the beta series models with 8 NVIDIA A100 GPUs, where they are all GPT-like models with around 7 billion parameters. It takes around three hours to train a beta series model and five hours to train a gemma one. Our codebase is adapted from the Alignment Handbook [63]. By comparing the validation loss on the test split (not used for later evaluation), we select the hyperparameter $\eta$ of both RPO (beta) and RPO (gemma) to be $0.005$. We list the remaining training configurations in Table 3, which are recommended by the Alignment Handbook.

| Configuration | Beta Series | Gemma Series |
|---|---|---|
| learning rate | 5.0e-7 | 5.0e-7 |
| learning scheduler type | cosine | cosine |
| warmup ratio | 1.0 | 1.0 |
| batch size | 128 | 128 |
| gradient accumulation | 2 | 16 |
| batch size per device | 8 | 1 |
| training epoch | 1 | 2 |
| $\beta$ | 0.01 | 0.05 |
| optimizer | adamw torch | adamw torch |
| seed | 42 | 42 |
| precision | bfloat16 | bfloat16 |

Table 3: Training configurations for beta series and gemma series models in this paper.

## F.2  Evaluation Details

**GPT-4 evaluation on the test split.**  We use the following prompts to guide GPT-4 to annotate the preferences among win, lose, and tie (we denote them by A, B, and C, respectively).

> **Prompts:** Please act as an impartial judge and evaluate the quality of the responses provided by two AI assistants to the user question displayed below. You should choose the assistant that follows the user's instructions and answers the user's question better. Your evaluation should consider factors such as the helpfulness, relevance, accuracy, depth, creativity, and level of detail of their responses. Begin your evaluation by comparing the two responses and provide a short explanation. Avoid any position biases and ensure that the order in which the responses were presented does not influence your decision. Do not allow the length of the responses to influence your evaluation. Do not favor certain names of the assistants. Be as objective as possible. After providing your explanation, output your final verdict by strictly following this format: [[A]] if assistant A is better, [[B]] if assistant B is better, and [[C]] for a tie. [Instruction] instruction [The Start of Assistant A's Answer] {*answer A*} [The End of Assistant A's Answer] [The Start of Assistant B's Answer] {*answer B*} [The End of Assistant B's Answer]

Here, we replace {*answer A*} and {*answer B*} with the answers of two models. Since GPT annotation has shown to prefer the answer in the first position [66], we randomly exchange the positions between two answers during the evaluation to ensure a fair comparison.

**Benchmark evaluation.**  We use the default configuration for the evaluations on MT-Bench[2] and AlpacaEval 2.0[3]. By default, the annotator of MT-Bench is the *latest version* of GPT-4. The default annotator and the competitor model are both GPT-4 (Preview 11/06). We only need to manually import the proper chat template that formats the training dataset, which are shown as follows.

---

[2]https://github.com/lm-sys/FastChat/tree/main/fastchat/llm_judge
[3]https://github.com/tatsu-lab/alpaca_eval/tree/main

> **Chat Template for Beta Series:** <|system|><|user|>
> {*instruction*}
> <|assistant|>

> **Chat Template for Gemma Series:** <bos> <|im_start|>user
> {*instruction*}<|im_end|>
> <|im_start|>assistant

### F.3 Additional Results on Experiments

In this section, we provide the additional results to show the performance gain for RPO (beta) in MT-Bench and RPO (gemma) in AlpacaEval 2.0. We report the pairwise win rates in Tables 4, 5, and 6 to analyze their performance gaps, where all the annotation configurations are the same in Table 2. Results show that RPO still exceeds DPO in the metric of the pairwise win rates on the benchmarks for both beta series and gemma series.

| win rate (%) | RPO (beta) | Ref. (beta) | DPO (beta) |
|---|---|---|---|
| RPO (beta) | 50.00 | **83.75** | **57.81** |
| Ref. (beta) | 16.25 | 50.00 | 21.25 |
| DPO (beta) | 78.75 | 42.19 | 50.00 |

Table 4: Pairwise win rates (left vs. right) for beta series models on MT-Benchmark.

| win rate (%) | RPO (beta) | Ref. (beta) | DPO (beta) |
|---|---|---|---|
| RPO(beta) | 50.00 | **80.13** | **52.02** |
| Ref.(beta) | 19.87 | 50.00 | 20.61 |
| DPO (beta) | 47.98 | 79.39 | 50.00 |

Table 5: Pairwise win rates (left vs. right) for gemma series models on AlpacaEval 2.0.

| win rate (%) | RPO (beta) | Ref. (beta) | DPO (beta) |
|---|---|---|---|
| RPO (beta) | 50.00 | **64.93** | **51.33** |
| Ref. (beta) | 35.07 | 50.00 | 36.44 |
| DPO (beta) | 48.67 | 64.56 | 50.00 |

Table 6: Pairwise Length-Control (LC) win rates (left vs. right) for gemma series models on AlpacaEval 2.0.

## G Experiments on Math, Reasoning, and Coding Tasks

### G.1 Experimental Details

To provide a more comprehensive analysis of the trained LLM, we introduce more benchmarks on the math, reasoning, and coding tasks for evaluations. Specifically, we choose the Grade School Math 8K (GSM8K), AI2 Reasoning Challenge (ARC), and Mostly Basic Python Programming (MBPP) to measure math, reasoning, and coding abilities, respectively. In this section, we focus on the gemma series for the experiments. We do not use chain-of-thought or few shots in all the benchmarks. We compare the greedy decoding result (pass @1) on the MBPP benchmark.

| Model Name | GSM8K (%) | ARC | | MBPP (Pass @1) | |
|---|---|---|---|---|---|
| | | Easy (%) | Challenge (%) | Normal (%) | Plus (%) |
| RPO | **49.9** | **79.1** | 49.8 | 54.2 | **46.3** |
| DPO | 45.3 | 75.7 | **50.0** | 54.2 | 43.9 |
| Ref. | 45.4 | 75.0 | 45.8 | 50.3 | 44.2 |
| `zephyr-gemma-7b` | 47.3 | 77.6 | 48.6 | **54.5** | 44.7 |

Table 8: Results on GSM8K, ARC, and MBPP. Here, `zephyr-gemma-7b` is the officially released models trained by DPO and Ref. denotes the reference model `zephyr-7b-gemma-sft` used for our training. RPO and DPO are trained with the OpenRLHF codebase [27] and we average the SFT loss regularizer in RPO by the number of tokens of the chosen response. We do not use chain-of-thought or few shots in all the benchmarks. We compare the greedy decoding result (pass @1) for MBPP.

Here we use the OpenRLHF codebase [27] to implement a new variant of RPO, where the SFT loss regularizer is averaged by the number of tokens of the chosen labels, that is, $(\log \pi_\theta(a_{\text{cho}}|x))/|a_{\text{cho}}|$. Such a variant balances the weight of the averaged SFT loss regularizer between the shorter chosen response and the longer one. We set the coefficient for the SFT loss regularizer as $0.2$. We use 8 NVIDIA A100 GPUs for the training and evaluation. The remaining hyperparameters are in Table 7.

| Configuration | Gemma Series |
|---|---|
| learning rate | 5.0e-7 |
| learning scheduler type | cosine with a minimum learning rate |
| batch size | 128 |
| gradient accumulation | 8 |
| batch size per device | 2 |
| training epoch | 2 |
| $\beta$ | 0.5 |
| optimizer | adamw torch |
| seed | 42 |
| precision | bfloat16 |

Table 7: Training configurations for DPO and RPO for the experiments in Appendix G.

## G.2 Experimental Results

Table 8 demonstrates that our proposed method still outperforms or performs equally to the vanilla DPO on these benchmarks of math, reasoning, and coding, which verifies the effectiveness of our proposed method.

