# OpenReview forum: "Provably Mitigating Overoptimization in RLHF: Your SFT Loss is Implicitly an Adversarial Regularizer"
_NeurIPS.cc/2024/Conference — NeurIPS 2024 poster_

### Official Review · Reviewer_1DUS · 2024-07-08

**Soundness:** 3
**Presentation:** 3
**Contribution:** 3
**Rating:** 7
**Confidence:** 4

**Summary:**

This paper aims to address the issue of over-optimization in RLHF. The authors introduce a method named RPO which concurrently minimizes the maximum likelihood estimation of the loss alongside a reward penalty term. Not only do the authors demonstrate that the proposed method is sample-efficient, but they also outline a straightforward yet effective implementation strategy. Experimental results underscore the efficiency of the method.

**Strengths:**

1.	The research introduces a novel RLHF method specifically designed to tackle the issue of over-optimization.

2.	Despite its simplicity in implementation, the method proves to be highly effective.

3.	The authors support their proposed method with thorough theoretical analysis, convincingly demonstrating that it benefits from finite-sample convergence guarantees.

**Weaknesses:**

1.	Additional experiments across a broader range of scenarios are required to more comprehensively demonstrate the method's efficiency.

2.	The current experiments are limited to evaluations using GPT and log probability, which do not offer intuitive insights into the over-optimization problem. In other words, it remains unclear whether the observed improvements in performance truly indicate a mitigation of the over-optimization issue. A more detailed analysis, perhaps focusing on rewards, could provide the necessary clarity.

**Questions:**

1.	The SFT loss appears similar to the commonly used PTX loss in [1]. Could you please elucidate their relationships and distinctions?

2.	How is the true reward (human reward) depicted in Figure 1 on the left derived?

3.	How does the proposed method address scenarios involving multiple reward models?

[1] Stanford alpaca: An instruction-following llama model.

**Limitations:**

Yes

---

> ### Author Rebuttal · Authors · 2024-08-07
>
> **Q1: Additional experiments are required to more comprehensively demonstrate the method's efficiency**
>
> **A1:** Please refer to the **General Response**.
>
> **Q2: The experiments are limited to evaluations using GPT and log probability. It remains unclear whether the observed improvements in performance truly indicate a mitigation of the over-optimization issue.**
>
> **A2:** Thanks for the question! First, we would like to point out that the actions and their chosen probabilities **can be** interpreted as a proxy of analyzing the underlying (estimated) reward model $\widehat{r}$ [1] due to the representation $\pi_{\widehat{r}}(a|x)\propto\pi^{\mathrm{ref}}(a|x)\exp(\beta^{-1}\widehat{r}(x,a))$. Analyzing the (log) probabilities of the actions can be used to detect the mitigation of over-optimization, because according to the representation, an overestimated reward of a poor action would result in a higher probablity of choosing this action, and would also cause a decay in the probability of choosing other better actions (since the probabilities are normalized to $1$).
>
> To further showcase the ability of RPO to address overoptimization (through the lense of probability), consider the following theoretical example with only actions [2] where we can track everything clearly. It has three actions $\{a, b, c\}$ with $R^\star(a) = 1, R^\star(b)=0.5,R^\star(c)=0$. The reference policy $\pi^{\mathrm{ref}}(a)=\pi^{\mathrm{ref}}(b)=0.4, \pi^{\mathrm{ref}}(c)=0.1$, and the dataset consists of one data point $\mathcal{D} = (a,b,1)$ (meaning action $a$ is preferred in the data). Then an ideally solved DPO objective would be $\pi_{\mathrm{DPO}}$ as long as $\pi^{\mathrm{DPO}}(b)=0$, and the value of $\pi^{\mathrm{DPO}}(a)$ can be arbitrarily chosen in $[0,1]$. Thus a possible solution to DPO would be $\pi^{\mathrm{DPO}}(a)=0.5,\pi^{\mathrm{DPO}}(b)=0$, and by the normalizing condition $\pi^{\mathrm{DPO}}(c)=0.5$, which is undesirable since the action $c$ has reward $R^{\star}(c)=0$. In contrast, solving the RPO objective would additionally require the maximization of $\pi_{\mathrm{RPO}}(a)$ due to the SFT regularization term, and thus the solution is shifted towards $\pi_{\mathrm{RPO}}(a)=1, \pi_{\mathrm{RPO}}(b)=\pi_{\mathrm{RPO}}(c)=0$, which is better than the DPO policy. Thus, RPO is able to prevent overoptimization towards poor actions that are less covered by the dataset (action $c$ here), therefore resulting in a better policy.
>
> **Q3: Could you please elucidate the relationships and distinctions between PTX loss in [2] and your SFT loss?**
>
> **A3:** Thanks for the question! The original PTX loss is an imitation loss calculated on the pretraining data. In contrast, the SFT loss in the RPO objective is an imitation loss calculated on the RLHF dataset. In more specific, our experiments use this SFT loss to imitate the chosen responses in the RLHF dataset. Thus the relationship is that they are both imitation loss which aims to mimic certain data distribution. The distinction is that they are calculated on different data sources. Moreover, the SFT loss in the RPO objective naturally comes from our theoretical algorithm and provably serves as an important regularization term to mitigate overoptimization in offline RLHF. We would make this comparison clearer in the revision.
>
> **Q4: How is the true reward (human reward) depicted in Figure 1 on the left derived?**
>
> **A4:** Thanks for pointing this out! Figure 1 (left) is an illustrative figure to showcase the mechanism behind overoptimization as a consequence of distributional shifts. The rewards therein does not correspond to actual human rewards and is plotted for illustrative purposes. Meanwhile, as we demonstrated in the answer of **Q2**, RPO can effectively address overoptimization depicted in Figure 1 (left) where data coverage is not sufficient. We will make this clearer in the revision.
>
> **Q5: How does the proposed method address scenarios with multiple reward models?**
>
> **A5:** Thanks for raising this interesting question! How our methods can address the overoptimization issue in this scenario depends on the specific learning target in the face of multiple objects, e.g., [4, 5, 6] and more references therein.
>
> For instance, when the goal is to find the optimal policy that maximizes a linear scalarized reward model [4, 6], the idea of RPO suggests to use the linearization of the multiple reward models to be learned as the regularizer in the object (3.2), which roughly give the object
> $$\max_{\pi}\min_{r^1\in\mathcal{R}^1,\cdots,r^m\in\mathcal{R}^m}\eta\mathbb{E}_{a^1\sim\pi,a^0\sim\pi^{\mathrm{base}}}[\mathbf{w}^\top\mathbf{r}(x,a^1)-\mathbf{w}^\top\mathbf{r}(x,a^0)]-\beta\mathrm{KL}(\pi\|\|\pi^{\mathrm{ref}}) + \mathcal{L}\_{\mathcal{D}^1,\cdots,\mathcal{D}^m}(\mathbf{r}).
> $$
> Inspired by our theory, such an algorithm can find the optimal policy as long as each of the responses in the data can cover the target policy in terms of the linearized reward (see Assumption 5.2, where the test function $r$ is replaced by the scalarization of the multiple rewards), thus overcoming the issue of overoptimization in offline RLHF. We leave the study of RPO for such multiple reward models as our future work.
>
> **References:**
>
> [1] Rafailov, Rafael, et al. "Direct preference optimization: Your language model is secretly a reward model." NeurIPS 36 (2024).
>
> [2] Xu, Shusheng, et al. "Is DPO Superior to PPO for LLM Alignment? A Comprehensive Study." 41th ICML.
>
> [3] Stanford alpaca: An instruction-following llama model.
>
> [4] Zhou, Zhanhui, et al. "Beyond one-preference-for-all: Multi-objective direct preference optimization." ArXiv:2310.03708 (2023).
>
> [5] Chakraborty, Souradip, et al. "MaxMin-RLHF: Towards equitable alignment of large language models with diverse human preferences." ArXiv:2402.08925 (2024).
>
> [6] Yang, Rui, et al. "Rewards-in-Context: Multi-objective Alignment of Foundation Models with Dynamic Preference Adjustment." 41th ICML.

---

> > ### Comment · Reviewer_1DUS · 2024-08-10
> >
> > Thank you for your response. I will keep my score positive.

---

> > > ### Author Response · Authors · 2024-08-11
> > > **Reply to the Official Comment by Reviewer 1DUS**
> > >
> > > Dear Reviewer 1DUS,
> > >
> > > Thank you for your review and support. We will incorporate your valuable suggestions into our paper as we revise it based on the feedback from all reviewers. Your comments greatly assist us in strengthening the overall quality of our work.
> > >
> > > Best regards, Authors

---

### Official Review · Reviewer_9VCJ · 2024-07-09

**Soundness:** 3
**Presentation:** 3
**Contribution:** 3
**Rating:** 6
**Confidence:** 5

**Summary:**

The paper introduces the concept of RPO, which combines DPO loss with SFT loss. This approach aims to align the policy with human preferences while simultaneously imitating a baseline distribution, effectively mitigating overoptimization. Empirical results from experiments with LLMs demonstrate that RPO outperforms traditional DPO methods, showcasing the practical applicability of the proposed algorithm.

**Strengths:**

The paper provides a robust theoretical framework for addressing overoptimization in RLHF. By identifying the source of misalignment as distributional shift and uncertainty, it offers a principled approach to the problem

The algorithm includes a reward penalty term to prevent the policy from exploiting spurious high proxy rewards, resulting in provable sample efficiency under partial coverage conditions

The paper provides empirical evidence demonstrating that RPO improves performance compared to DPO baselines in aligning LLMs. This practical validation strengthens the theoretical claims made in the study

**Weaknesses:**

The theoretical guarantees provided by the algorithm rely on specific conditions, such as partial coverage. These conditions might not always hold in practical scenarios, potentially limiting the generalizability of the results​.

The SFT loss + DPO seems very intuitive.

**Questions:**

It is noticed believed that math problems are not very suitable with vanilla DPO. Some practitioners found similar algorithms should help the performance. Have you checked the reasoning benchmarks and the proposed algorithm?

**Limitations:**

Yes.

---

> ### Author Rebuttal · Authors · 2024-08-07
>
> **Q1: The theoretical guarantees provided by the algorithm rely on specific conditions such as *partial coverage*, which might not always hold in practical scenarios, potentially limiting the generalizability of the results.**
>
> **A1:** Thanks for raising the question. We would like to comment that all the assumptions we imposed to obtain the theoretical guarantees are quite standard in the RL theory literature. Actually, our theory features a *minimal assumption* on the data distribution (the partial coverage assumption) thanks to our algorithm design. Similar kinds of data assumptions also appear in recent theoretical works on RLHF, e.g., [1, 2, 3].
>
> To explain more, the partial coverage assumption (Assumption 5.2) only requires the dataset to cover the policy $\pi$ to compete. As is shown by [4], the partial-coverage-style data assumption is the minimal assumption such that provably sample-efficient offline RL is possible. That being said, this assumption is actually a weak condition in terms of the data distribution, especially in comparison with the stronger notion of uniform coverage [5, 6, 7] where the offline dataset needs to cover all possible policies. Moreover, our theory works in the regime of general function approximation (instead of linear regimes [1, 3]), which also exhibits its generality.
>
> But still, when going from theory to practice, the implementation of our algorithm RPO itself does not require the knowledge of these assumptions. It can be directly applied to handle overoptimization in RLHF for real-world problems. This is demonstrated by the effectiveness of RPO in LLM fine-tuning shown in the paper.
>
> **Q2: The proposed algorithm (SFT loss + DPO loss) seems very intuitive.**
>
> **A2:** Yes! the resulting algorithm does look very intuitive. This is actually an *advantage* of our algorithm design. Typically, to *provably* address overoptimization in offline RLHF with general function approximations, the theoretical algorithm would rely on solving complicated non-convex constrained optimizations over certain confidence regions [1, 2], which is prohibitive to scale to practice like LLMs without modifications or adaptations.
>
> In contrast, our proposed theoretical object (Algorithm 1) naturally induces the simple but equivalent form of RPO (Algorithm 2) after delicate mathematical deductions (see Section 4). That is, it suffices to add an SFT loss to the preference optimization loss to implement the theoretical algorithm in an equivalent manner. Therefore, the simple form of RPO as well as the theoretical guarantees it enjoys is one of our main contributions. Also, our experimental results demonstrate its effectiveness despite its simple form.
>
> Finally, we notice that adding an SFT-style loss as a regularizer in RLHF object is becoming more and more popular, which has been adopted by the fine-tuning of Llama 3.1 [8] (see Section 4.1.4). Given that, our work also serves as a theoretical foundation for such an effective practice in large-scale RLHF.
>
> **Q3: About the reasoning benchmarks of the proposed algorithm**
>
> **A3:** Please refer to the **General Response**.
>
> **References:**
>
> [1] Zhu, B., Jordan, M., & Jiao, J. (2023, July). Principled reinforcement learning with human feedback from pairwise or k-wise comparisons. In International Conference on Machine Learning (pp. 43037-43067). PMLR.
>
> [2] Zhan, Wenhao, et al. "Provable Offline Preference-Based Reinforcement Learning." The Twelfth International Conference on Learning Representations.
>
> [3] Xiong, W., Dong, H., Ye, C., Wang, Z., Zhong, H., Ji, H., ... & Zhang, T. (2024). Iterative preference learning from human feedback: Bridging theory and practice for rlhf under kl-constraint. In Forty-first International Conference on Machine Learning.
>
>
> [4] Jin, Ying, Zhuoran Yang, and Zhaoran Wang. "Is pessimism provably efficient for offline rl?." International Conference on Machine Learning. PMLR, 2021.
>
> [5] Munos, Rémi. "Error bounds for approximate policy iteration." ICML. Vol. 3. 2003.
>
> [6] Chen, Jinglin, and Nan Jiang. "Information-theoretic considerations in batch reinforcement learning." International Conference on Machine Learning. PMLR, 2019.
>
> [7] Xie, Tengyang, and Nan Jiang. "Q* approximation schemes for batch reinforcement learning: A theoretical comparison." Conference on Uncertainty in Artificial Intelligence. PMLR, 2020.
>
> [8] Dubey, Abhimanyu, et al. "The Llama 3 Herd of Models." arXiv preprint arXiv:2407.21783 (2024).

---

> > ### Comment · Reviewer_9VCJ · 2024-08-12
> >
> > Thank you for your response. I have revised the score.

---

> > > ### Author Response · Authors · 2024-08-12
> > > **Reply to the Official Comment by Reviewer 9VCJ**
> > >
> > > Dear Reviewer 9VCJ,
> > >
> > > Thank you for your review and support. We will incorporate your valuable suggestions into our paper as we revise it based on the feedback from all reviewers. Your comments greatly assist us in strengthening the overall quality of our work.
> > >
> > > Best regards, Authors

---

### Official Review · Reviewer_JyGr · 2024-07-14

**Soundness:** 3
**Presentation:** 3
**Contribution:** 3
**Rating:** 6
**Confidence:** 4

**Summary:**

The paper "Provably Mitigating Overoptimization in RLHF" addresses the issue of overoptimization in aligning large language models (LLMs) with human preferences using reinforcement learning from human feedback (RLHF).

The main contributions include:
1. Identification of Overoptimization Source: The paper identifies the source of reward overoptimization as a distributional shift and uncertainty in learning human preferences.

2. Theoretical Algorithm Proposal: It proposes a theoretical algorithm that minimizes the maximum likelihood estimation of the loss and a reward penalty term to mitigate overoptimization, ensuring provable sample efficiency.

3. Practical Implementation: The algorithm is reformed into an easy-to-implement objective combining preference optimization and supervised learning loss, named Regularized Preference Optimization (RPO), demonstrating improved performance in aligning LLMs compared to existing methods​

**Strengths:**

This paper not only provides rigorous analysis but also has solid experiments to solve the overoptimization problem for DPO.

**Weaknesses:**

The partial coverage condition lacks discussions since now it's a pair over (\pi,\pi^{base}), which is different from the traditional coverage condition. For example, for the linear case, $C_{\mu_D}$ would approximately become
$$
\mathbb{E}_{x,a^1\sim\pi^*,a^2\sim\pi^{pref}}\sqrt{(\phi(x,a^1) - \phi(x,a^0))^{\top} \Sigma_D^{-1} (\phi(x,a^1) - \phi(x,a^0))},
$$
where $\pi^{pref}$ means the distribution of the chosen samples. We know that $\Sigma_D$ is composed of pairs of chosen and unpreferred samples, but the $(\phi(x,a^1) - \phi(x,a^0))$ is there pair of optimal policy and the policy represents the chosen samples. Hence, if we want to compete with a policy $\pi$ better than $\pi^{chosen}$, how can the direction of $(\pi,\pi^{chosen})$ be covered by $(\pi^{unprefered},\pi^{chosen})$?

**Questions:**

1. I just wonder about the extension to online RLHF. For online RL, based on the optimism principle, it seems that then the objective should be subtracted from the SFT loss, which obliviates the wish to avoid overoptimization. So how to balance the exploration and avoiding overoptimization for the online setting?

2. What is the additional computational complexity brought by the gradient of SFT loss? Besides, the author doesn't mention how to approximate the gradient of SFT loss since there are expectations.

**Limitations:**

See weaknesses.

---

> ### Author Rebuttal · Authors · 2024-08-07
>
> **Q1: The partial coverage condition lacks discussions since now it's a pair over $(\pi,\pi^{base})$, which is different from the traditional coverage condition. Hence, if we want to compete with a policy $\pi$ better than $\pi^{\mathrm{chosen}}$, how can the direction of $(\pi, \pi^{\mathrm{chosen}})$ be covered by $(\pi^{\mathrm{unchosen}},\pi^{\mathrm{chosen}})$?**
>
> **A1:** Thanks for raising this question! and we appreciate your suggestion to include more detailed discussions and explanations on this partial coverage coefficient. Here, we briefly explain this condition and address your concern on its rationality.
>
> Essentially, a sufficient condition to make this partial coverage condition (Assumption 5.2) hold is that the distribution of the offline dataset, which is $\mu_\mathcal{D}$, can well cover the joint distribution of $(a^1, a^0)\sim (\pi, \pi^{\mathrm{base}})$. We can focus on $\pi^{\mathrm{base}} = \pi^{\mathrm{chosen}}$ as we adopted in the paper.
>
> First, we would like to clarify that the offline dataset distribution $\mu_\mathcal{D}$ is not simply $(a^1, a^0)\sim(\pi^{\mathrm{unchosen}}, \pi^{\mathrm{chosen}})$ as understood by the reviewer, since according to our definition (see Section 2) whether $a^1$ or $a^0$ is chosen is random and is determined by $y \in\{0, 1\}$ obeying the BT model. Thus, $(a^1, a^0)\sim \mu_\mathcal{D}$ can be interpreted as a mixture of $(\pi^{\mathrm{unchosen}}, \pi^{\mathrm{chosen}})$ and $(\pi^{\mathrm{chosen}}, \pi^{\mathrm{unchosen}})$. This mixture probability would not be too small as long as the quality of $(a^1, a^0)$ does not vary too much, i.e., both of them are possible to be chosen, which is the case in practice. As a result, in the offline data distribution $(a^1,a^0)\sim \mu_{\mathcal{D}}$, both $a^1$ and $a^0$ partly comes from the chosen distribution $\pi^{\mathrm{chosen}}$.
>
> Then in order for $\mu_\mathcal{D}$ to cover the joint distribution of $(a^1, a^0)\sim (\pi, \pi^{\mathrm{base}})$, it suffices to argue that $\pi^{\mathrm{chosen}}$ can cover the target policy $\pi$, which is then reduced back to the traditional coverage condition. Thus our assumption essentially requires that $\pi^{\mathrm{chosen}}$ well covers and only needs to cover the target policy $\pi$. This coincides with the spirit of the minimal data assumption in offline RL theory, i.e., the so-called partial coverage condition.
>
> We will make this clearer in the revision of our paper.
>
> **Q2: For online RL, based on the optimism principle, it seems that then the objective should be subtracted from the SFT loss, which obliviates the wish to avoid overoptimization. So how to balance the exploration and avoiding overoptimization for the online setting?**
>
> **A2:** Thank you for pointing this out! Actually online RLHF is a different theoretical setup than offline RLHF. The goal of *regret minimization* in online learning does not face the problem of overoptimization, because the data are not precollected but are collected and updated interactively. This in turn requires exploration and needs the algorithm to be optimistic.
>
> When our technique is applied to online RLHF, it does induce a similar SFT loss subtracted from the preference optimization loss, but the baseline policy $\pi^{\mathrm{base}}$ in the SFT loss needs to be chosen carefully and is not necessarily $\pi^{\mathrm{chosen}}$ as we considered in the offline setup. Possible candidates for the baseline policy could be the LLM at the previous iteration (serve as the reference policy for the current iteration). In this way, the data distribution of the actions (responses generated by the currently learned LLM) can be gradually shifted towards that of the optimal actions.
>
> Thus, theoretically, we only need to subtract a properly designed SFT loss for online RLHF in terms of regret minimization. For practical situations where one might still need to handle the overoptimization issue after online data collection, we conjecture that the optimal way is to use optimism during the online data collection stage (subtract a properly designed SFT loss) and perform pessimism after all the data have been collected (add SFT loss as RPO). Still, online RLHF is beyond the scope of this paper. We leave this interesting question of addressing overoptimization in online RLHF to future work.
>
> **Q3: About the computational complexity and the implementations of the SFT loss gradient**
>
> **A3:** According to the paragraph **Practical implementation** in Section 6, RPO adds an additional SFT loss (the log probability of the chosen labels in the preference dataset) on the original DPO loss, where the SFT loss is a intermediate quantity in the calculation of DPO loss. Hence, our proposed method will not incur any additional computation overhead compared with the vanilla DPO. As for the justification of the approximation of the SFT loss, we use the linear property of the expectation to show that the population form of the RPO loss $\mathcal{L}_ {\text{RPO}}$ can be rewritten as
> $$
> \mathcal{L}_ {\text{RPO}}(\theta) = \mathbb{E}_ {(x,a_{\text{cho}},a_{\text{rej}})\sim \mu_ {\mathcal{D}}}\Bigl[-\log \pi_\theta(a_ {\text{cho}}\mid x) +\sigma \bigl(\hat r_ \theta(x,a_ {\text{cho}}) -  \hat r_ \theta(x,a_ {\text{rej}})\bigr)\Bigr],
> $$
> where we denote by $\hat r_\theta(x,a)=\beta\cdot\log(\pi_\theta(a\mid x))/\log(\pi_{\text{ref}}(a\mid x))$ and denote $\mu_{\mathcal{D}}$ as the population distribution of the preference dataset $\mathcal{D}$. It suggests that we only need to sample a mini-batch $\mathcal{D}_ {\text{mini}}$ from $\mu_{\mathcal{D}}$ (or equivalently sample from $\mathcal{D}$) and calculate the gradient w.r.t. $\theta$ on
> $$
> \mathbb{E}_ {(x,a_{\text{cho}},a_{\text{rej}})\sim {\mathcal{D}_ {\text{mini}}}}\Bigl[-\log \pi_\theta(a_ {\text{cho}}\mid x) +\sigma \bigl(\hat r_ \theta(x,a_ {\text{cho}}) -  \hat r_ \theta(x,a_ {\text{rej}})\bigr)\Bigr],
> $$
> which approximates the gradient $\nabla_\theta\mathcal{L}_ {\text{RPO}}(\theta)$.

---

> > ### Comment · Reviewer_JyGr · 2024-08-09
> >
> > The authors have addressed my question, and I choose to maintain my score.

---

> > > ### Author Response · Authors · 2024-08-11
> > > **Reply to the Official Comment by Reviewer JyGr**
> > >
> > > Dear Reviewer JyGr,
> > >
> > > Thank you for your review and support. We will incorporate your valuable suggestions into our paper as we revise it based on the feedback from all reviewers. Your comments greatly assist us in strengthening the overall quality of our work.
> > >
> > > Best regards, Authors

---

### Author Rebuttal · Authors · 2024-08-07

**General Response:**

We thank all the reviewers for their time and effort reviewing our paper and we appreciate all your support of our work! We have responded to each of you detailedly.

Here we provide a general response to **Q3** of **Reviewer 9VCJ** and **Q1** of **Reviewer 1DUS** about more experimental evaluations of the proposed algorithm RPO. To this end, we use extra benchmarks on the math, reasoning, and coding tasks to showcase the effectiveness of our method. Please refer to the PDF document attached to this response for detailed results. Thank you!

---

### Decision · Program_Chairs · 2024-09-25

**Decision:**

Accept (poster)

**Comment:**

The paper provides an interesting method to mitigate overoptimization in LLMs. The reviewers all agree with the contribution and novelty of the results. I recommend acceptance.